

# Comparison of particle number size distribution trends in ground measurements and climate models

Ville Leinonen[1], Harri Kokkola[2], Taina Yli-Juuti[1], Tero Mielonen[2], Thomas Kühn[1,2], Tuomo Nieminen[3], Simo Heikkinen[1], Tuuli Miinalainen[1], Tommi Bergman[4,5], Ken Carslaw[6], Stefano Decesari[7], Markus Fiebig[8], Tareq Hussein[9,10], Niku Kivekäs[11], Markku Kulmala[9], Ari Leskinen[2,1], Andreas Massling[12], Nikos Mihalopoulos[13], Jane P. Mulcahy[14], Steffen M. Noe[15], Twan van Noije[5], Fiona M. O'Connor[14], Colin O'Dowd[16], Dirk Olivie[17], Jakob B. Pernov[12,18], Tuukka Petäjä[9], Øyvind Seland[17], Michael Schulz[17], Catherine E. Scott[6], Henrik Skov[12], Erik Swietlicki[19], Thomas Tuch[20], Alfred Wiedensohler[20], Annele Virtanen[1], and Santtu Mikkonen[1,21]

[1]Department of Applied Physics, University of Eastern Finland, Kuopio, Finland
[2]Finnish Meteorological Institute, Kuopio, Finland
[3]Institute for Atmospheric and Earth System Research, University of Helsinki, Helsinki, Finland
[4]Climate System Research, Finnish Meteorological Institute, Helsinki, Finland
[5]Royal Netherlands Meteorological Institute, De Bilt, Netherlands
[6]Institute for Climate and Atmospheric Science, School of Earth and Environment, University of Leeds, Leeds, UK
[7]Institute of Atmospheric and Climate Sciences (ISAC) of the National Research Council of Italy (CNR), Bologna, Italy.
[8]Department of Atmospheric and Climate Research, NILU-Norwegian Institute for Air Research, Kjeller, Norway
[9]Institute for Atmospheric and Earth System Research/Physics, Faculty of Science, University of Helsinki, Helsinki, Finland
[10]Department of Physics, the University of Jordan, Amman 11942, Jordan
[11]Finnish Meteorological Institute, Helsinki, Finland
[12]Department of Environmental Science, iClimate, Aarhus University, Denmark
[13]Environmental Chemical Processes Laboratory (ECPL), Chemistry Department, University of Crete, Heraklion, Crete, Greece
[14]Met Office Hadley Centre, Exeter, United Kingdom
[15]Institute of Forestry and Engineering, Estonian University of Life Sciences, Tartu, Estonia
[16]School of Natural Sciences and Ryan Institutes Centre for Climate and Air Pollution Studies, National University of Ireland Galway, Galway, Ireland
[17]Norwegian Meteorological Institute, Oslo, Norway
[18]Now at Extreme Environments Research Laboratory, École Polytechnique fédérale de Lausanne, 1951 Sion, Switzerland
[19]Division of Nuclear Physics, Physics Department, Lund University, Lund, Sweden
[20]Leibniz Institute for Tropospheric Research, Leipzig, Germany
[21]Department of Environmental and Biological Sciences, University of Eastern Finland, Kuopio, Finland

*Correspondence to*: Annele Virtanen (*annele.virtanen@uef.fi*) and Ville Leinonen (*ville.j.leinonen@uef.fi*)

## Abstract

Despite a large number of studies, the effect of aerosols has the largest uncertainty in global climate model radiative forcing estimates. There have been studies of aerosol optical properties in climate models, but the effects of particle number size distribution need a more thorough inspection. We investigated the trends and seasonality of particle number concentrations in different sizes in total of for 21 measurement sites in Europe and Arctic. For 13 of those, with longer measurement time series, we compared the field observations with the results from five climate models, namely EC-Earth3, ECHAM-M7, ECHAM-





SALSA, NorESM1.2, and UKESM1. This is the first extensive comparison of detailed aerosol size distribution trends between in-situ observations from Europe and five earth system models (ESM). We found that the trends of particle number concentrations were mostly consistent and decreasing in both, measurements and models. However, for many sites, climate models showed weaker decreasing trends than the measurements. Seasonal variability in measured number concentrations,

quantified by the ratio between maximum and minimum monthly number concentration, were typically stronger in northern measurement sites compared to other locations. Models had large differences in their seasonal representation, and they can be roughly divided into two categories. For EC-Earth and NorESM, the seasonal cycle was relatively similar for all sites, for others, the pattern of seasonality varied between northern and southern sites. In addition, the variability in concentrations across sites varied between models, some having relatively similar concentrations for all sites, whereas others showing clear

differences in concentrations between remote and urban sites. To conclude, although anthropogenic mass emissions are harmonized in models, trends in different sized particles vary among the model due to assumptions in emission sizes and differences in how models treat size dependent aerosol processes. The inter-model variability was largest in the accumulation mode, i.e. sizes which have implications for aerosol-cloud interactions.

## 1 Introduction

Atmospheric aerosols form one of the most important components that cool the climate, counteracting heating by increased greenhouse gas concentrations (Forster et al., 2021). Aerosol-radiation-interactions (ARI) and aerosol-cloud-interactions (ACI) greatly depend on particle concentration, size distribution and chemical properties, and ACI altogether their ability to activate to cloud droplets. On the other hand, the ability of large-scale climate models to predict the aerosol direct and indirect radiative forcing depends mainly on their ability to describe the spatial and temporal distribution and characteristics of the

atmospheric aerosol population. Especially the strength of cooling due to ACI depends on the number concentration of particles large enough to activate to cloud droplets (Dusek et al., 2006). The ability of global-scale models to reproduce the trends of these particles is important for reproducing the changes in aerosol radiative forcing and further, diagnosing the radiative forcing from anthropogenic emissions. Improvement of aerosol radiative forcing estimates, which are still the most uncertain part of total radiative forcing estimates (Forster et al., 2021), would improve the estimate of total radiative forcing, the climate

sensitivity and future climate change (Myhre et al., 2013).

It has been projected that both air pollution and climate change mitigation measures will lead to decreased emissions of anthropogenic aerosols (Smith and Bond, 2014). In addition, a global warming driven temperature increase affects the emissions of biogenic volatile compounds and formation of secondary organic aerosol, and through that concentrations and size distribution characteristics of atmospheric aerosols (Arneth et al., 2010; Hellén et al., 2018; Mielonen et al., 2012;

Paasonen et al., 2013; Peñuelas and Staudt, 2010; Yli-Juuti et al., 2021). Atmospheric aerosols have already undergone significant changes caused by tightened air pollution control measures. For example, Hamed et al. (2010) showed a clear reduction in aerosol concentrations in Melpitz, Germany between 1996 and 2006, which was associated with sulphur dioxide



(SO₂) emission reductions in Europe. Several other studies have reported significant changes in the atmospheric aerosol population showing clear negative trends in particle concentrations in different size ranges (Mikkonen et al., 2020; Sun et al., 2020) as well as for total number concentration and mass (Asmi et al., 2013; Collaud Coen et al., 2013). The change in aerosol optical properties has been consistent with these observations, with aerosol optical depth showing a decreasing trend over Europe and Arctic (Breider et al., 2017; Collaud Coen et al., 2013, 2020; Schmale et al., 2022).

Observations of particle number concentrations and their optical properties, as well as radiation measurements, help to constrain how well climate models simulate the climate effects of aerosols. Storelvmo et al. (2018) showed that models from the 5ᵗʰ Coupled Model Intercomparison Project (CMIP5) are not reproducing the observed trends in incoming surface solar radiation (SSR). Moseid et al. (2020) showed that the same holds also for the CMIP6 models. Since SSR is affected by aerosol extinction and cloud cover, the analysis of Moseid et al. (2020) indicated that the discrepancy between models and observations was related, at least partly. to erroneous aerosol and aerosol precursor emission inventories.. Mortier et al. (2020) studied the trends of particle optical properties and found that the trends were mostly decreasing for measured optical parameters, and climate models were mainly showing relatively similar trends. However, models usually underestimate aerosol optical parameters such as optical thickness and scattering (Gliß et al., 2021). These findings indicate a need for further analysis comparing observed trends of the aerosol population with trends from global models.

Comparing in-situ aerosol observations with global model outputs is not straightforward due to differing temporal and spatial scales represented. In-situ measurements represent one point while a global scale model simulates average aerosol properties within a grid box, which can be on the order of 100 km in horizontal resolution and on the order of a few tens of meters in the vertical at the level of the observations. The scale differences make a one-to-one comparison of models and observations at a specific time incoherent unless the in-situ observation is a good representative of the mean value of the model grid box area. On the other hand, the proximity of the observation site to emission sources, changes in local wind speed and direction, and the dynamics of the boundary layer can cause large fluctuations at the measurement site, which cannot be captured with the coarse resolution of global models and may not be representative for a larger area. However, using long time series and a large number of observational sites allows for bridging the gap between the scales (Schutgens et al., 2017). In addition, collocating the observations and model data in time allows for a closer comparison of the two (Schutgens et al., 2016).

In this study, we perform an aerosol number size distribution trend analysis for observations from 21 European and Arctic sites, analyse the trends of particle mode properties (number concentration, geometric mean diameter, and geometric standard deviation) and compare 13 sites with simulations from the recent past from five climate models. In addition, we compare the yearly seasonal cycle representation of the models to the measured seasonal cycle.

## 2 Data and methods

We investigated the characteristics of particle number size distributions by separating the size distribution into log-normal modes (nucleation, Aitken, and accumulation mode). We analysed the number concentration, geometric mean diameter, and





geometric standard deviation, and their trends for sites representing polar (Villum, Zeppelin), Arctic remote (Pallas, Värriö), rural (Birkenes II, Hohenpeißenberg, Hyytiälä, Järvselja, Melpitz, San Pietro Capofiume), rural regional background (K-Puszta, Neuglobsow, Waldhof, Vavihill), urban (Annaberg-Buchholz, Helsinki, Leipzig, Puijo), coastal remote (Mace Head, Finokalia), and high altitude (Schauinsland) environments. Finally, to evaluate how well current climate models can reproduce the observed aerosol physical trends and seasonal variability, we compared observations from 13 selected sites with results

from five different climate models. The selection criterion was for the measurement sites to provide at least 7 years of observational data between 2001 and 2014.

Measurement data sets differ in the reported aerosol size range and time resolution. Furthermore, the climate modelling data used are averages over the grid boxes containing the coordinates of the respective measurement sites. It is therefore not straightforward to compare measurement data of different locations or to compare measured and modelled data. In order to

make such comparisons meaningful, the data must be adjusted and modified in a consistent manner. In this section, we go through the data modification process used and explain and verify the chosen approaches and methods.

Daily and monthly averages of number size distribution parameters are used in the trend analysis. We are using the Dynamic Linear Model (DLM) (Petris et al., 2009) to evaluate short-term changes in trends (based on the data of daily averages) and Sen-Theil estimators for long term trend estimation (monthly averages) and comparing with the modelled trends of climate

models (monthly averages). Seasonality of observed and climate model output number concentrations of each aerosol distribution mode are compared with seasonality metrics introduced in Rose et al. (2021) using monthly data.

## 2.1 Data from measurement sites

### 2.1.1 Measurement sites

Data sets used in this study are partly the same as in the study of Nieminen et al. (2018) and are supplemented by newer data

from The Aerosol, Clouds and Trace Gases Research Infrastructure (ACTRIS) sites (www.actris.eu) and SmartSmear (https://smear.avaa.csc.fi/). From ACTRIS sites, we have also included new sites that were not included in Nieminen et al. (2018) (Annaberg-Buchholz, Birkenes II, Leipzig, Neuglobsow, Puijo, Schauinsland, and Waldhof), and expanded the data length by including recent years that were missing in Nieminen et al. (2018). In addition, data from Villum Research Station at Station Nord (Villum) and some recent years' data from Puijo and San Pietro Capofiume were received directly from the

research groups operating the sites.

In this study, we have used only long-term observations (minimum 6 years of measurement data) of particle number size distributions. The length of the data sets and corresponding data coverage varies between the sites (see Fig. S1), being between 59.6 and 98.4% of the days of the measurement period of each site. The measurement sites used in this study are listed in Table 1. For model comparison, we have included only those sites that have at least 7 years of a common time period with the model

simulations (2001-2014) and sufficient data coverage (i.e., coverage > 50% of days). In Table 1, the sites are presented in two



separate lists: The first list is showing the sites that are used both in trend analysis and comparisons of observational and model trends and the second list is showing sites that were used only in trend analysis.

Site environment classification is adapted from Nieminen et al. (2018) for those sites that were included in their study. For other sites, we have used classifications from the literature (Sun et al., (2020) for German sites, Leskinen et al. (2012) for

Puijo, Schmale et al. (2018) for Vavihill, and Nguyen et al. (2016) for Villum) for environment classification and adjusted their classification according to Nieminen et al. (2018).

It should be noted that there is a significant variation in the detected size ranges of the measurement instruments between the sites and within one site over the analysed time period (see Table 1). For those sites where the size range has varied over the investigated time period, we have limited the analysis only to the size range that has been measured over the whole analysis

period. This size range is site-specific to maximize the amount of data in each site. We have interpolated the data to site-specific, common size resolution, i.e., the size bins of size distribution data were same for the whole time period.

**Table 1 Information of measurement sites used in this study. Site name, site environment type, coordinates, and altitude in meters above sea level, time period, and size range (rounded to nearest nm for minimum size and nearest 10 nm for maximum size) covered.**

**Sites in both trend analysis and model comparison**

| Site name | Environment | Location | Altitude (m.a.s.l.) | Time period | Size range (nm) |
|---|---|---|---|---|---|
| Helsinki, Finland | Urban | 60°12'N 24°58'E | 26 | 2005–2018 | 3–1000 |
| Hohenpeißenberg, Germany | Rural | 47°48'N 11°1'E | 988 | 2008–2018 | 13–800 |
| Hyytiälä, Finland | Rural | 61°51'N 24°17'E | 181 | 1996–2018 | 3–500 |
| K-Puszta, Hungary | Rural, reg. bg. | 46°58'N 19°33'E | 125 | 2008–2018 | 7–710 |
| Puijo, Finland | Semi-urban | 62°55'N 27°40'E | 306 | 2006–2015 | 10–500 |
| Mace Head, Ireland | Remote | 53°12'N 9°48'W | 10 | 2005–2012 | 21–500 |
| Melpitz, Germany | Rural | 51°32'N 12°54'E | 87 | 2008–2018 | 5–800 |
| Pallas, Finland | Remote | 67°58'N 24°7'E | 565 | 2008–2018 | 7–500 |
| San Pietro Capofiume, Italy | Rural | 44°39'N 11°37'E | 11 | 2002–2015 | 3–630 |
| Schauinsland, Germany | High-altitude, reg. bg. | 47°55'N 7°55'E | 1205 | 2006–2018 | 10–600 |
| Vavihill, Sweden | Rural, reg. bg. | 56°1'N 13°9'E | 172 | 2001–2017 | 3–860 |
| Värriö, Finland | Remote | 67°45'N 29°36'E | 390 | 1998–2018 | 8–400 |
| Zeppelin, Norway | Polar | 78°56'N 11°53'E | 474 | 2008–2018 | 10–800 |

**Sites in trend analysis**

| Site name | Environment | Location | Altitude (m.a.s.l.) | Time period | Size range (nm) |
|---|---|---|---|---|---|
| Annaberg-Buchholz, Germany | Urban bg. | 50°34'N 12°59'E | 545 | 2012–2018 | 10–800 |
| Birkenes II, Norway | Rural | 58°23'N 8°15'E | 219 | 2010–2018 | 10–550 |
| Finokalia, Greece | Remote | 35°23'N 25°40'E | 235 | 2011–2018 | 9–760 |



| Järvselja, Estonia | Rural | 58°16'N 27°16'E | 36 | 2012–2017 | 3–10000 |
|---|---|---|---|---|---|
| Leipzig, Germany | Urban bg. | 51°21'N 12°26'E | 118 | 2010–2018 | 10–800 |
| Neuglobsow, Germany | Rural, reg. bg. | 53°8'N 13°2'E | 70 | 2012–2018 | 10–800 |
| Waldhof, Germany | Rural, reg. bg. | 52°48'N 10°45'E | 75 | 2009–2018 | 10–800 |
| Villum, Greenland | Polar | 81°36'N 16°40'W | 30 | 2010–2018 | 9–910 |

### 2.1.2 Quality checking of measurement data

To study separately the evolution of particle number concentration and size in each mode of the size distribution, we fitted three lognormal modes (nucleation, Aitken, and accumulation) to the measured data. Before fitting the modes, we first performed a visual examination of the size distribution time series to detect clear errors in the data that could affect the results of the fitting process, e.g., the absence of some modes in the fit due to problems in the data. For example, if a substantial fraction (over 20% of the size bins) of the number-size distribution was not measured during a specific size distribution measurement, the whole distribution was removed.

### 2.1.3 Fitting of log-normal modes to particle number size distributions

To investigate the trends in particle number size distributions, multimodal log-normal size distributions were fitted to the measured data and the trend analysis was performed on the mode parameters. We fitted one to three modes for each particle size distribution using an automatic mode-fitting algorithm (Hussein et al., 2005). Briefly, the algorithm fits a combination of one to three log-normal distributions to the particle number size distribution data, separately for each time step at each location. The algorithm assumes three log-normal modes as a starting point and reduces automatically the number of modes if any of the overlapping conditions for modes is true (for more details, see Hussein et al., 2005). For each mode, the algorithm returns three parameters: geometric mean diameter, $D_p$, geometric variance, $\sigma_p^2$, and mode number concentration, $N$.

For each fit, a quality check was performed. Firstly, we checked that the number concentrations of the fitted modes were reasonable. We used measured size bin diameters as a limit and omitted those cases where the geometric mean diameter of mode was smaller than the smallest size bin or larger than the largest size bin from the analysis. To avoid possible overestimation of the number concentration of the modes, we assigned the number concentration of the missing or removed modes to be zero, with missing geometric diameter and geometric standard deviation.

We noticed that in cases where the smallest size bin of the measured size distribution had a high number concentration, the mode fitting algorithm did not perform well and, instead, fitted a nucleation mode that had an unreasonably high number concentration and often also a geometric mean diameter outside of the measured size range. The reason for this was that the geometric mean diameter of the nucleation mode was smaller than the smallest detected size of the instrument, especially in cases where this size was relatively large. For the nucleation mode, this limitation removed a median of 17.8% of the fitted nucleation modes amongst all sites, ranging from 0 to 41.1% (Mace Head) between sites. For the accumulation mode, a similar





phenomenon was observed, resulting in high number concentrations for large diameters near the largest detected size, although was less likely (<0.1 % of the fitted accumulation modes).

The fitted modes were sorted into three categories, nucleation, Aitken, and accumulation mode, based on their geometric mean diameter. In the case of three fitted modes, the modes were arranged based on geometric mean diameters, with one mode

always being assigned to each category. In cases with one or two fitted modes, the assignment was primarily based on the mean diameter of the mode. Here a cut-off of 20 nm was used for the fitted geometric mean diameter to distinguish between nucleation and Aitken modes, and a cut-off of 100 nm was used to distinguish between Aitken and accumulation modes. Sometimes two fitted modes both fell within the same category. In such cases, the mode was assigned to categories based on the diameter. If both modes had diameters between 20 and 100 nm (1.7% of the cases), the mode with a diameter further from

those cut-off points was assigned to be Aitken mode, and the other mode, depending on its diameter, was assigned to be nucleation or accumulation mode. If both modes had diameters larger than 100 nm (0.4% of the cases), the mode with the larger diameter was assigned to accumulation mode and the mode with the smaller diameter to Aitken mode. There were no cases where both modes had diameters below 20 nm.

As a result of the fitting, we categorized modes for each measured size distribution. The time resolution of the measured size

distributions, and consequently the fitted modes, varied between sites and ranged from 3 min to 60 min. For further analysis, we calculated daily means for each fitted mode parameter (i.e., $N$, $D_p$, and $\sigma$). For the mean to be calculated, there had to be at least 50 % of measurements available for a day (i.e. 12 hours of data).

We further studied when a fraction of the different modes was missing at each site. The absence of a fitted mode at certain time points was dependent on the mode (nucleation, Aitken, or accumulation) and site. The absence was most probably caused

by low concentrations of particles within the mode size range. The Aitken mode was most often present, and the nucleation mode was most often missing. Daily percentages of mode occurrence, i.e., in which fraction of measurements a certain mode was fitted for each day, for each measurement site are presented in Table 1 and Figures S2 and S3. For Aitken and accumulation modes, the mode occurrence was more than 80 % for most of the days in all sites and was close to 100% (i.e., mode was fitted for every observation) in most of the sites. For the nucleation mode, the mean mode occurrence was around 80 %; however,

there are sites where the occurrence was much lower. This can be due to limitations of size distribution measurements for nucleation mode particles (size range starting from > 10 nm) or lack of nucleation mode particles e.g., due to meteorological or emission-related reasons. The latter is suggested by observations of nucleation occurrence in Fig. S2: urban sites had a reasonably high representation also in the nucleation mode, whereas remote sites had days during which the nucleation mode was fitted for only a few or even zero measurement points per day. More detailed figure about representation as a function of

month and hour of day is presented in Figure S3. There were differences in nucleation mode representation during a day and during a year, nucleation mode most often being fitted after midday. However, the patterns were not uniform for all the sites, and especially for Mace Head, the lower limit of the detected particle size most probably affected the results.





To conclude, the absence of modes and taking the daily mean of observed modes did not affect Aitken and accumulation modes drastically. Results for nucleation mode number concentrations are more uncertain compared to results for the other modes which should be kept in mind when interpreting the results.

For comparison between climate models and observations, we also computed monthly means (trend analysis) and seasonal medians (SeasC calculation) of the fitted log-normal modes to the observational data described above. As global model results were monthly means, the same time resolution was also applied for the mode data. Monthly means of the measured data were calculated using the daily-averaged data, with the limitation that at least five daily mean values per month were required. This limitation removed only two months from the entire dataset, in addition to the months that were completely missing from the observational data. Seasonal means and seasonal medians were computed using monthly means with at least two monthly means per season being required.

### 2.1.4 Remapping measurement data sets for comparison with climate models

As shown later in the results section, the mean diameters of the fitted modes are larger than the corresponding diameters/bins used in climate models. This might affect the model-observation comparison results, especially for the nucleation mode, where the relative difference between the diameters of fitted modes and model modes is largest. Therefore, we calculated separate representations of the measurement data, which are more directly comparable to the model results: for the modal and sectional aerosol schemes, the measurement data were re-binned using the model limits. For comparison, with the Sectional Aerosol module for Large Scale Applications (SALSA), the measured size bins with a mean geometrical diameter of 3 to 7.7 nm were assigned to the nucleation mode. This size range corresponds to the limits of the smallest size bin in a SALSA (Kokkola et al., 2018). Measured size bins from 7.7 to 50 nm (corresponding to the second and third smallest size bins in SALSA) were assigned to the Aitken mode, and from 50 to 700 nm (fourth to sixth smallest size bins in SALSA) to the accumulation mode. In the modal representation for comparison with the modal models, the corresponding size limits were 3 to 10 nm for nucleation, 10 to 100 nm for Aitken, and 100 to 1000 nm for accumulation mode. As can be seen from Table 1, the corresponding diameter range of each mode category from the models is not fully captured by the measurements at every site. If measurements were covering only a part of the model's diameter range, that part has been used as a representative mode from measurements if there are at least three size bins of measurement data available. This limitation was used because the number concentrations from one or two bins have a large variance, resulting in very uncertain trends. If there were fewer bins or no measurement data available, the corresponding nucleation mode is not represented in the results section. For Aitken and accumulation modes, there were always enough data to calculate representative modes, even though the accumulation mode is not always measured up to the diameter of 1000 nm.



**Table 2 Daily median and mean coverage and the standard deviation of the coverage of the fitted nucleation, Aitken, and accumulation modes at measurement sites during the whole measurement time series.**

| Site | Nucleation modes fitted (% of observations/day) | | | Aitken modes fitted (% of observations/day) | | | Accumulation modes fitted (% of observations/day) | | |
|---|---|---|---|---|---|---|---|---|---|
| | median | mean | std. dev. | median | mean | std. dev. | median | mean | std. dev. |
| Annaberg-Buchholz | 70.8 | 63.6 | 26.4 | 100.0 | 99.3 | 2.2 | 100.0 | 95.4 | 7.3 |
| Birkenes II | 29.2 | 31.1 | 22.7 | 100.0 | 99.3 | 3.4 | 100.0 | 93.2 | 10.6 |
| Finokalia | 45.8 | 47.1 | 23.7 | 100.0 | 99.6 | 2.2 | 100.0 | 98.6 | 4.8 |
| Helsinki | 91.6 | 88.1 | 11.4 | 100.0 | 98.6 | 3.1 | 93.1 | 89.0 | 11.7 |
| Hohenpeißenberg | 54.2 | 55.1 | 22.1 | 100.0 | 99.4 | 2.8 | 100.0 | 95.9 | 8.4 |
| Hyytiälä | 72.9 | 70.5 | 18.9 | 100.0 | 98.8 | 3.8 | 99.3 | 95.7 | 7.5 |
| Järvselja | 46.0 | 47.7 | 19.2 | 99.0 | 96.8 | 5.4 | 96.8 | 90.5 | 13.6 |
| K-Puszta | 60.0 | 59.5 | 20.9 | 100.0 | 99.3 | 2.3 | 100.0 | 97.8 | 5.0 |
| Leipzig | 69.6 | 66.7 | 19.0 | 100.0 | 99.0 | 2.7 | 100.0 | 95.7 | 7.3 |
| Mace Head | 20.8 | 28.5 | 28.1 | 100.0 | 100.0 | 0.2 | 100.0 | 98.6 | 4.4 |
| Melpitz | 78.3 | 74.7 | 18.3 | 100.0 | 98.7 | 3.3 | 100.0 | 96.7 | 6.9 |
| Neuglobsow | 41.7 | 43.0 | 21.6 | 100.0 | 99.5 | 2.3 | 100.0 | 97.0 | 6.6 |
| Pallas | 52.9 | 51.6 | 23.5 | 100.0 | 96.5 | 8.7 | 100.0 | 94.6 | 8.6 |
| Puijo | 55.0 | 54.5 | 16.8 | 100.0 | 98.1 | 3.6 | 97.5 | 93.1 | 9.7 |
| San Pietro Capofiume | 78.5 | 76.5 | 15.9 | 99.3 | 97.9 | 3.4 | 97.2 | 93.8 | 8.5 |
| Schauinsland | 58.3 | 57.6 | 22.1 | 100.0 | 99.3 | 2.9 | 100.0 | 95.9 | 8.1 |
| Värriö | 36.1 | 37.5 | 21.0 | 100.0 | 97.9 | 5.8 | 100.0 | 97.3 | 5.8 |
| Vavihill | 82.6 | 77.7 | 18.7 | 100.0 | 99.0 | 4.0 | 100.0 | 95.1 | 11.2 |
| Villum | 33.7 | 36.4 | 22.1 | 100.0 | 97.4 | 7.5 | 100.0 | 96.6 | 9.3 |
| Waldhof | 66.7 | 65.9 | 21.1 | 100.0 | 99.2 | 2.9 | 100.0 | 96.6 | 7.0 |
| Zeppelin | 37.5 | 39.4 | 26.8 | 100.0 | 96.9 | 8.3 | 100.0 | 94.7 | 11.4 |

## 2.2 Data from climate models

We used climate model data from EC-Earth3-AerChem (van Noije et al., 2021), The Norwegian Earth System Model NorESM1.2 (Kirkevåg et al., 2018), and UK's Earth System Model UKESM1 (Sellar et al., 2019) which participated in model simulations carried out within the European Union funded project CRESCENDO (Coordinated Research in Earth Systems and Climate: Experiments, Knowledge, Dissemination and Outreach). CRESCENDO simulations ran from the year 2000 to 2014 except for NorESM1.2, which ran from 2001 to 2014. All the models were run in atmosphere-only configuration with sea surface temperatures and sea ice concentrations prescribed as in the Atmospheric Model Intercomparison Project (AMIP)





simulation of the Coupled Model Intercomparison Project Phase 6 (CMIP6). The climate models provided monthly values for the aerosol number size distribution, making the data useful for comparison against observations. In addition, we ran two configurations of the global aerosol-chemistry-climate model ECHAM6.3-HAMMOZ2.3-MOZ1.0, one with the M7 modal

aerosol model (Tegen et al., 2019) and one with the sectional aerosol model SALSA (Kokkola et al., 2018). Specific features and the aerosol representation of each model is described in the following sections and summarized in Table 2.

From the global model calculations, we selected results for grid boxes containing the coordinates of the respective measurement sites and calculated the number concentrations of nucleation, Aitken, and accumulation mode particles. If both soluble and insoluble particle concentrations were provided for the mode, the sum of those has been used as the total number

concentration of that mode.

**Table 3 Summary of model setup, emissions, and aerosol microphysics in five climate models used in this study.**

### Model setup

| Model name | Description of size distribution | Horizontal resolution | Vertical resolution | Nudging |
|---|---|---|---|---|
| ECHAM-M7 | Seven log-normal modes, nucleation soluble, Aitken soluble, Aitken insoluble, accumulation soluble, accumulation insoluble, coarse soluble, coarse insoluble | T63 (~1.9° x 1.9°) | L47, top at 0.01 hPa | Era Interim |
| ECHAM-SALSA | 17 size sections in total, 10 soluble bins (3nm - 10 um in diameter), 7 insoluble bins (50nm - 10 um in diameter) | T63 (~1.9° x 1.9°) | L47, top at 0.01 hPa | Era Interim |
| EC-Earth3 | Seven lognormal modes, nucleation soluble, Aitken soluble, Aitken insoluble, accumulation soluble, accumulation insoluble, coarse soluble, coarse insoluble | IFS: TL255 (i.e., a spectral truncation at wavenumber 255 with a linear N128 reduced Gaussian grid, corresponding to a spacing of about 80 km), TM5: 2° × 3° (latitude × longitude) | IFS: L91, top at 0.01 hPa, TM5: L34, top at 0.1 hPa | |
| NorESM.2 | Twelve modes, based on mixed particles in nucleation, Aitken, accumulation and coarse size range with BC, OM, sulphate, dust and sea salt as core substrate. | 0.9° × 1.25° (latitude x longitude) | L30, top at approx 3 hPa | Era Interim |
| UKESM1 | Five lognormal modes, nucleation soluble, Aitken soluble, Aitken insoluble, accumulation soluble, coarse soluble | 1.25° × 1.88° (latitude × longitude) | L85, top at approx 85km | Era Interim |

### Emissions

| Model name | Sea salt | Dust | SO$_X$ | NO$_3$ |
|---|---|---|---|---|



| Model name | | | | |
|---|---|---|---|---|
| ECHAM-M7 | Online calculated based on Guelle et al. (2001) | Online calculated based on Tegen et al. (2002) with modifications described in Cheng et al. (2008) and Heinold et al. (2016) | Volcanic emissions: Carn 2017 (AeroCom Phase III; explosive and degassing emissions for the year 2010); Anthrop. & biomass: CMIP6 | N/A |
| ECHAM-SALSA | Same as ECHAM-M7 | Same as ECHAM-M7 | Same as ECHAM-M7 | N/A |
| EC-Earth3 | Online calculated based on Gong (2003) and Salter et al. (2015) | Online calculated based on Tegen et al. (2002). | Anthropogenic and biomass burning emissions of SOx from CMIP6; effusive volcanic emissions of SOx from Andres and Kasgnoc (1998). | N/A |
| NorESM1.2 | Salter et al. (2015) | Online calculated in the land-model, based on Zender et al. (2003) | Anthrop. & biomass: CMIP6, effusive volcanic: Dentener et al. (2006). | N/A |
| UKESM1 | (Gong, (2003) | Updated version of Woodward (2001) - see Mulcahy et al. (2020) for details. | Anthrop. (no SO2 from biomass burning in UKESM1): CMIP6 (Hoesly et al., 2018); effusive volcanic: (Dentener et al., 2006). | N/A |

| Model name | Organic aerosol (OA) | Black Carbon | Dimethyl sulphide (DMS) | NH$_3$ |
|---|---|---|---|---|
| ECHAM-M7 | Secondary OA (SOA) is 15% of prescribed natural terpene emissions at the surface Dentener et al. (2006); Anthrop. & biomass: CMIP6 | Anthrop. & biomass: CMIP6 | Online calculated using sea water concentrations from Lana et al. (2011); parameterisation with air-sea exchange from Nightingale et al. (2000) | N/A |
| ECHAM-SALSA | Same as ECHAM-M7 | Same as ECHAM-M7 | Same as ECHAM-M7 | N/A |
| EC-Earth3 | Anthropogenic and biomass burning emissions from CMIP6; biogenic emissions from MEGANv2.1 Sindelarova et al. (2014) for the year 2000. Marine organic emissions are not included. | Anthropogenic and biomass burning emissions from CMIP6; biogenic emissions from MEGANv2.1 Sindelarova et al. (2014) for the year 2000. | Oceanic DMS emissions were calculated online based on Lana et al. (2011) and Wanninkhof (2014). Terrestrial DMS emissions from soils and vegetation are prescribed following Spiro et al. (1992). | Anthropogenic and biomass burning emissions of NH3 from CMIP6; biogenic emissions of NH3 from soils under natural vegetation and oceanic emissions of NH3 from Bouwman et al. (1997). |
| NorESM1.2 | Natural emissions of particulate organic matter and volatile organic compounds for SOA as in Kirkevåg et al. (2018). Anthrop. & biomass: CMIP6. | Anthrop. & biomass: CMIP6. | Online calculated using sea water concentrations from Lana et al. (2011); parameterisation with air sea exchange from Nightingale et al. (2000). | N/A |





| | | | |
|---|---|---|---|
| UKESM1 | Natural marine emissions of POM follow Gantt et al. 2011, 2012;; UKESM1 has an interactive BVOC scheme which uses Pacifico et al. (2011) for isoprene; (Guenther et al. (1995) for monoterpene. Note only monoterpene sources currently feed into SOA formation, isoprene source not used in aerosol scheme - see Mulcahy et al. (2020); Anthrop. & biomass burning OC CMIP6 (Hoesly et al., 2018; van Marle et al., 2017). | Anthrop. And biomass burning: CMIP6 | Oceanic DMS emissions calculated online based on seawater DMS concentrations produced by the MEDUSA ocean biogeochemistry model (Yool et al., 2013) - this uses a modified version of the Anderson et al. (2001) - see Mulcahy et al. (2020); air sea emission flux is calculated using Liss and Merlivat (1986). | N/A |

**Aerosol microphysics**

| Model name | Nucleation mechanism | SOA formation |
|---|---|---|
| ECHAM-M7 | Ion-induced nucleation (Kazil et al., 2010) | SOA is assumed to condense immediately on existing aerosol particles and to have identical properties to primary organic aerosols |
| ECHAM-SALSA | Activation type nucleation (Sihto et al., 2006) | Same as ECHAM-M7 |
| EC-Earth3 | Riccobono et al. (2014) + Binary nucleation (Vehkamäki, 2002) | Bergman et al. (2022) |
| NorESM1.2 | Makkonen et al. (2014), Kirkevåg et al. (2018) | Kirkevåg et al. (2018) |
| UKESM1 | Binary homogeneous nucleation follows (Vehkamäki, (2002). There is currently no representation of boundary layer nucleation of new particles. | Simple oxidation of monoterpene produces a condensable secondary organic species which can condense onto pre-existing particles. |

### 2.2.1 EC-Earth3

The atmospheric component of the global climate model EC-Earth3-AerChem (van Noije et al., 2021) consists of a modified
version of the general circulation model used in the Integrated Forecasting System (IFS) cycle 36r4 from the European Centre
for Medium-Range Weather Forecasts (ECMWF), and the aerosol and chemistry model TM5. The IFS model version applied
in EC-Earth3-AerChem has a horizontal resolution of TL255 (i.e., a spectral truncation at wavenumber 255 with a linear N128
reduced Gaussian grid, corresponding to a spacing of about 80 km), and uses 91 hybrid sigma-pressure levels in the vertical
direction with a model top at 0.01 hPa. TM5 uses an atmospheric grid with a reduced resolution of $2° \times 3°$ (latitude × longitude)
and 34 vertical layers extending to ~ 0.1 hPa. The data exchange between the two model components is governed by the OASIS
coupler.



The aerosol scheme of TM5 is based on the modal aerosol microphysical scheme M7 from Vignati et al. (2004), which includes sulphate, black carbon, organic aerosols, sea salt and mineral dust. In TM5, the formation of secondary organic aerosols is described as in Bergman et al. (2022). The concentrations of ammonium, nitrate, and the aerosol water associated with

(ammonium) nitrate are calculated assuming equilibrium gas-particle partitioning. In the current model version, this equilibrium is calculated from the Equilibrium Simplified Aerosol Model (EQSAM, Metzger et al., 2002). The chemistry scheme of TM5 accounts for gas-phase, aqueous-phase, and heterogeneous chemistry (van Noije et al., 2021). The sources of mineral dust and sea salt, the oceanic source of DMS, and the production of nitrogen oxides by lighting are calculated online. Emissions from anthropogenic activities and open biomass burning are prescribed using data sets provided by CMIP6. All

other emissions are prescribed as documented in van Noije et al. (2021).

### 2.2.2 ECHAM-HAMMOZ

ECHAM-HAMMOZ (echam6.3-hammoz2.3-moz1.0) is a global aerosol-chemistry-climate model which consists of the atmospheric circulation model ECHAM (Stevens et al., 2013), the aerosol model HAM (Kokkola et al., 2018; Tegen et al., 2019), and the chemistry model MOZ (Schultz et al., 2018) not used in this study. The model solves atmospheric circulation

in three dimensions with spectral truncation of T63 which corresponds to approximately 1.9° × 1.9° horizontal resolution and uses 47 vertical layers extending to 0.01 hPa. The model includes the sectional aerosol model SALSA, which describes size distributions using 10 size bins between 3 nm – 10 μm in diameter, with externally mixed parallel size bins between 50 nm – 10 μm for treatment of particles consisting of insoluble material when they are emitted. The ECHAM-HAMMOZ also includes an option of using the modal aerosol model M7 which describes the aerosol size distribution with a superposition of seven log-

normal modes. Details of how aerosol processes are calculated in SALSA are described by Kokkola et al. (2018). The same details for M7 are described by Tegen et al. (2019).

Both model configurations (i.e., SALSA and M7) were set up according to the AeroCom (Aerosol Comparisons between Observations and Models) initiative phase III experiment setup. Anthropogenic aerosol emissions were according to Community Emissions Data System (CEDS; Hoesly et al., 2018), for biomass burning, we used Biomass Burning Emissions

for CMIP6 (BB4CMIP; van Marle et al., 2017). Dust, sea salt, and maritime DMS emissions are calculated online as a function of 10 m wind speed (see Tegen et al., 2019 and references therein). Atmospheric circulation (vorticity, divergence, and surface pressure) was nudged towards ERA-Interim reanalysis data (Berrisford et al., 2011) but temperature was allowed to evolve freely.

### 2.2.3 NorESM1.2

NorESM1.2 (Kirkevåg et al., 2018) is an earth-system model which consists of the atmospheric model CAM5.3-Oslo, the sea-ice model CICE4, the land model CLM4.5, and an updated version of the MICOM ocean model used in NorESM1 (Bentsen et al., 2013). CAM5.3-Oslo is based on CAM5.3 (Liu et al., 2016; Neale et al., 2012), but contains a different aerosol scheme





(OsloAero5.3), along with other small modifications. In this study, the model is run with a horizontal resolution of 0.9° × 1.25°, and 30 layers in the vertical (model top at around 3 hPa).

The aerosol scheme in NorESM1.2 describes aerosols using 12 separate modes, which can consist of sulphate, BC, OM (including SOA), sea-salt or dust (see Kirkevåg et al., 2018 for a detailed description), and its interaction with radiation and clouds. Emission strength of natural aerosol-precursors and aerosols such as dust, sea salt, primary marine organic matter, marine DMS, isoprene and monoterpenes are calculated interactively (Kirkevåg et al., 2018). The nucleation scheme for new particle formation used in NorESM1.2 is described in Makkonen et al. (2014). We have used the anthropogenic emissions

from Hoesly et al. (2018) and biomass burning emissions from van Marle et al. (2017). We prescribed sea-surface temperatures and sea-ice concentrations based on observations, and in the atmosphere, the horizontal wind (U and V) and surface pressure were nudged to 6-hourly ERA-Interim reanalysis data.

### 2.2.4 UKESM1

The United Kingdom Earth System Model (UKESM1) is described in detail by Sellar et al. (2019) and is built around the
Global Coupled 3.1 (GC3.1) configuration of the HadGEM3 (Hadley Centre Global Environment Model) physical climate model (Kuhlbrodt et al., 2018; Williams et al., 2018). UKESM1 additionally includes ocean and land biogeochemical processes and a stratospheric–tropospheric chemistry scheme (Archibald et al., 2020) implemented as part of the United Kingdom Chemistry and Aerosol (UKCA) model. In the simulations performed for the CRESCENDO project, UKESM1 was set to operate at a horizontal resolution of 1.25° × 1.88° (latitude × longitude), with 85 vertical levels.

The representation of aerosols within UKESM1 is described and evaluated by Mulcahy et al. (2020); UKESM1 employs the modal version of the Global Model of Aerosol Processes (GLOMAP) two-moment aerosol microphysics scheme (Mann et al., 2010). The aerosol number size distribution is represented by soluble nucleation, Aitken, accumulation, and coarse (diameter > 1000 nm) modes, and an additional insoluble Aitken mode. The above modes are used to carry information about sulphate, black carbon, particulate organic matter and sea salt whilst mineral dust is treated using the separate sectional scheme of
Woodward (2001). In UKESM1, there is no parameterised new particle formation scheme applied in the boundary layer. Anthropogenic emissions of aerosols are prescribed from the CMIP6 inventories: SO2 and anthropogenic BC and OC are taken from the Community Emissions Data System (CEDS; Hoesly et al., 2018), and biomass burning emissions are from van Marle et al. (2017). UKESM1 interactively simulates emissions of marine DMS, biogenic volatile organic compounds (BVOCs) and primary marine organic aerosol (Sellar et al., 2019).

### 2.3 Data analysis methods

#### 2.3.1 Observational short-term trends: Dynamic Linear Model (DLM)

We used the Dynamic linear model (DLM) for determining the short-term variation in trends, i.e. transient changes in the (long term) trend in timescales of some months to some years, of different measured mode parameters in the daily data set, we have





used the Dynamic linear model (DLM) (Durbin and Koopman, 2012; Laine, 2020; Petris et al., 2009). The main advantage of

DLM compared to many other non-parametric and parametric trend estimation methods is that DLM can also detect a non-monotonic trend and the seasonality of the time series can be estimated simultaneously with the trend.

DLM explains the measured variability of the time series $y_t$ of the mode parameter $(N, D_p, \text{or } \sigma)$ with three components. Firstly, the level component $\mu_t$ that is locally linear, but the trend $\alpha_t$ can change during the measured period. Secondly, a seasonality component $\gamma_t$ captures the seasonal pattern of the time series. Thirdly, a residual component $\eta_t$ that uses an

autoregressive model (AR(1), $\rho$), accounts for autoregression of the time series, i.e. dependence of the daily measurement on that from its previous day, and finally a normally distributed random noise components $\varepsilon_t, \varepsilon_{level,t}, \varepsilon_{trend,t}, \varepsilon_{seas,t}$, and $\varepsilon_{AR,t}$ which are related to uncertainties in each component. For each observation $y_t$ at time $t$, the DLM model used in this study is given by:

$$y_t = \mu_t + \gamma_t + \eta_t + \varepsilon_t, t = 1, \dots, T$$


$$\mu_t = \mu_{t-1} + \alpha_t + \varepsilon_{level,t,}$$

$$\alpha_t = \alpha_{t-1} + \varepsilon_{trend,t}$$

$$\sum_{i=0}^{11} \gamma_{t-i} = \varepsilon_{seas,t,}$$

$$\eta_t = \rho\eta_{t-1} + \varepsilon_{AR,t,}$$

where $\varepsilon_t \sim N(0, \sigma_t)$, $\varepsilon_{level,t} \sim N(0, \sigma_{level}^2)$, $\varepsilon_{trend} \sim N(0, \sigma_{trend}^2)$, $\varepsilon_{seas} \sim N(0, \sigma_{seas}^2)$, and $\varepsilon_{AR}^2 \sim N(0, \sigma_{AR}^2)$. We have used $\rho = $

0.4 as a value for AR (1)-coefficient in all model fittings. The initial value of the level has been set to be the yearly mean of the first year. Calculation of the DLM model has been done in the Matlab environment (MATLAB, 2019) using the DLM Matlab Toolbox (Laine et al., 2014).

As the applied DLM formulation assumes normally distributed data, we used log10-transformation for mode number concentrations. If number concentration was zero (i.e. no fitted modes were available for that day), we used a value of one as

a number concentration for that day to avoid problems with log10-transformation. For mode diameter and geometric standard deviation, no transformations were applied. We investigated the residuals $\varepsilon_t$ after the model fitting and in most cases, the assumptions of the model are sufficiently fulfilled, with the distribution of the residuals being close to a normal distribution. Before interpreting the level and the trend of the number concentration of each mode, we have transformed the level $\mu_t$ and trend $\alpha_t$ back to the original scale by using the exponential back-transformation.

**2.3.2 Long-term linear trends: Sen-Theil estimator**

Long-term trends of measured mode parameters in the dataset were estimated using the Sen-Theil estimator (Sen, 1968; Theil, 1950). The Sen-Theil estimator is a non-parametric method to estimate a linear trend. The advantages of the Sen-Theil estimator compared to more common linear regression methods are that it does not assume normality of the data and it is more





robust to outliers. Compared to the more complex DLM model, the Sen-Theil estimator also works with a lower number of
data points, which is one reason we used it in the model comparison.

Trend estimation was performed using the *TheilSen* function from *openair* package in the R environment (Carslaw and
Ropkins, 2012; R Core Team, 2021). The calculation of 95% confidence intervals is based on bootstrap method (Kunsch,
1989). Trend estimation was done for whole year data (monthly averages) and seasonal data (seasonal data of monthly
averages). Before trend estimation for the whole year data set, time series was de-seasonalized with seasonal trend
decomposition using loess and autocorrelation for consecutive months was taken into account when calculating the uncertainty
of the trend estimates. Seasons have been defined to be 3 months each, winter consisting of December-February, spring March-
May, summer June-August, and autumn September-November. In the trend estimation for observational data sets (Section
3.1), we have used all months available from each site. In all comparisons of observations and models (Section 3.2), we used
only those months that were available from the measurement sites.

We have used relative change (%/year) as the main parameter for comparing results. Relative change has been calculated for
the Sen-Theil estimator and confidence intervals by using the option slope-percent. The function uses the fitted value of a first
observation as a reference for calculating relative change (Carslaw and Ropkins, 2012).

### 2.3.3 Magnitude and pattern of seasonality

The seasonality of particle number concentration and its magnitude is highly varying between different measurement sites,
depending on e.g. latitude and environment type of site (Asmi et al., 2013; Rose et al., 2021) and the mode studied. Similarly,
parameters such as CCN number concentrations and NPF frequency have a seasonal cycle (Asmi et al., 2011; Nieminen et al.,
2018). Seasonality of the optical properties in models has been studied (Gliß et al., 2021) but for particle number concentrations
we are not aware of studies that compare measurements and models based on long-term data sets.

We compared the seasonality of number concentrations in models and measurements by studying modes separately. We used
two variables, the Normalized Interquartile Range (NIQR) and SeasC (Rose et al., 2021) to compare seasonality between
models and measurements. When calculating these seasonal parameters from measurements and model results, we included
only those months for which the measurement and model data were available. We calculated NIQR and SeasC separately for
each year to also assess the distribution of values in the studied period.

NIQR, defined as $NIQR = \frac{3rd\ Quartile - 1st\ Quartile}{Median}$, describes the interquartile range of observations for one year. NIQR was
calculated using monthly averages of concentrations, with at least 10 monthly averages needed to be available. The calculation
of NIQR is slightly different from Rose et al. (2021) who used daily values calculating NIQR. As we had only monthly averages
from model data, daily values could not be used. Based on the measurement data, we checked whether the time resolution
would change the NIQR values, by comparing NIQR values calculated from daily and monthly averages. We found that the
NIQR calculated from daily averages were usually higher, sometimes as much as twice the one calculated from monthly





averages. Therefore, NIQR values presented in this study are not comparable to values presented in Rose et al. (2021) but only

between the different data sets in this study, or others calculated from monthly averages.

SeasC is the ratio of maximum and minimum of seasonal median values, calculated separately for each year and mode in each

data set. It was calculated by first taking the seasonal averages for each season. For calculating the seasonal median, at least

two monthly means from the season were required. Then, if we were able to calculate all the seasonal medians for the year,

SeasC was calculated as the ratio of the maximum and minimum of those seasonal medians.

In general, both SeasC and NIQR describe the distribution of number concentrations within one year. SeasC is focusing more

on utmost values, minimum and maximum of seasonal medians, whereas NIQR is focusing on values closer to the yearly

median. Neither SeasC nor NIQR considers when the maximum and minimum in number concentrations are achieved. Though

the seasonal cycle of the measured and modelled number concentrations might be opposite to each other, the difference in

SeasC or NIQR values can be small when comparing measurements and model data.

To assess whether the seasonal maximums and minimums have similarities between measurements and models, we have

calculated the seasonal averages, selected the seasons that have most often had seasonal maximum and minimum during

measured time period and evaluated how modelled results are corresponding to the measurements.

## 3 Results

### 3.1 Observational number size distribution characteristics and trends in daily in situ measurement data sets

We investigated the mode characteristics (number concentration $N$, geometric mean diameter $D_p$, and geometric standard

deviation $\sigma$) for nucleation, Aitken and accumulation modes for 21 European and Arctic sites representing Polar (Villum,

Zeppelin), arctic remote (Pallas, Värriö), rural (Birkenes II, Hohenpeißenberg, Hyytiälä, Järvselja, Melpitz, San Pietro

Capofiume), rural regional background (K-Puszta, Neuglobsow, Waldhof, Vavihill), urban (Annaberg-Buchholz, Helsinki,

Leipzig, Puijo), coastal remote (Mace Head, Finokalia) and high altitude (Schauinsland) environments. Median values and

interquartile ranges for different mode parameters for the sites over the analysis period are shown in Fig. 1 (and for different

seasons in Fig. S4-S6). Figure 1 shows a large variation in $N$s between the sites. As expected, the Arctic and other remote sites

had the lowest concentrations overall while urban sites and central European sites had the highest concentrations, especially

for the nucleation and Aitken modes. Generally, $N$ were higher for southern compared to northern sites. Partially the

differences between southern and northern sites could be explained by the site environmental types: more polluted site types

were typically found in south. However, the concentrations for southern sites were higher also within site classes. For the

accumulation mode, the highest $N$ were found in more polluted rural sites in central Europe, K-Puszta and San Pietro

Capofiume. These results are in line with previous results for number concentrations, such as found by Rose et al. (2021).

For $D_p$ and $\sigma$, results were not as distinctive for different environments. Standard deviations $\sigma$ were highest for nucleation

modes, and lowest for accumulation modes without clear differences between site environmental types.





Coastal sites Finokalia and Mace Head showed the largest $D_p$ in Aitken and accumulation modes, while Birkenes II (rural) and Mace Head showed the largest $D_p$s in nucleation mode. Järvselja (rural) had the lowest $D_p$ in all modes. One aspect that could explain some of the differences in $D_p$ between sites is the lower limit of the detected size range in the measurements. The lower value of the smallest detectable size might increase the probability that the $D_p$ of fitted nucleation mode is smaller.

For example, for Mace Head site the lowest measured size bin is around 21 nm, affecting the $D_p$ of the fitted nucleation mode. The lowest detected size may also affect the fitted Aitken mode diameter. However, for Finokalia and Järvselja, the measured size range could not completely explain observed high and low $D_p$ of the nucleation, respectively. This was tested by using a minimum size of ~10 nm for those sites that have measured < 10 nm particles and calculating the mode parameters as in Fig. 1. For this test, $D_p$ were calculated using ~10 nm as lowest size in Finokalia were close to diameters using the original lowest

size in Fig. 1. Geometric mean diameters in Järvselja were increasing by some nanometres but were still lowest among all sites, except in nucleation mode, where Villum then had the lowest $D_p$.

To investigate the effect of measurement size range on mode fitting, we studied the dependence of $D_p$ and minimum size bin measured amongst all sites. Spearman's rank correlation between $D_p$ and lowest size bin amongst sites was positive, 0.67 for nucleation, 0.01 for Aitken, and 0.21 for accumulation mode indicating strongest dependence for nucleation modes, and only

a minor dependence for accumulation modes. Thus, especially for nucleation modes, the lowest detectable size is related to the lower $D_p$ in Fig. 1.

Results for $D_p$ are somewhat different compared to what has been observed in Rose et al. (2021). They reported that mode diameters for Aitken and accumulation modes were smallest for urban sites (32 ± 11 nm and 122 ± 37 nm), followed by mountain (39 ± 9 and 142 ± 25), polar (42 ± 14 and 149 ± 37), and continental (51 ± 13 and 174 ± 29) sites. In our results,

most urban sites had a smaller Aitken mode $D_p$ compared to most of the rural continental sites, with the most notable exceptions from this tendency being Puijo and Järvselja. However, otherwise, the differences between site types reported by Rose et al. (2021) were not observed in our study. In general, the $D_p$ were smaller in our study, however, the rural sites in our study and continental sites in Rose et al. (2021) have accumulation mode diameters close to each other. Rose et al. (2021) studied only particles ranging from 20 to 500 nm and year 2016 or 2017, depending on the site. They also had a larger number

of sites considered. In our analysis, the analysed particle size range has in particular affected the mean diameters since at least part of the 20-30 nm particles were fitted into the nucleation mode, whereas in Rose et al. (2021), those were included to the Aitken mode. As a result, the fitted Aitken modes in our study had slightly larger $D_p$ compared to fitting only Aitken and accumulation mode.

It is worth noting that the fitted modes and their diameters were mostly larger than what is usually assumed in climate models.

Fitted nucleation modes had mean diameters from above 10 nm (Järvselja) to around 20 nm (Mace Head), while the upper limit of nucleation mode in sectional (7 nm) and modal (10 nm) model representations are below all the medians of fitted mean diameters to the observational data.



**Figure 1. Summary of mode parameters (number concentration $N$, geometric mean diameter $D_p$, and geometric standard deviation $\sigma$) for the measurement sites. The median values are marked with dots and interquartile ranges (25% and 75%) with whiskers for different mode parameters in fitted modes.**

To investigate the short-term trends at different measurement sites over the analysed time periods, we used DLM analysis as described in Section 2.3.1. To demonstrate the characteristics of a DLM trend fit, Aitken mode $N$s and their estimated level for the Mace Head site are shown in Fig. 2. Aitken mode $N$ at Mace Head were selected as an example because there is a substantially large increase in number concentration during the measured period, which is also seen in Fig. 3 showing the estimated trend in Aitken mode for all sites. The trend in Mace Head given by DLM (red line in Fig. 2) were temporarily over 10%/year. It must be noted that the concentrations at Mace Head were quite low compared to many other sites, and the variation of average $N$ in Aitken modes between days was relatively large, ranging from 50 to 3000 particles $cm^{-3}$. The number of high





concentration days (here denoted as $> 500$ particles $cm^{-3}$ on average) increased towards the year 2010 and has been decreasing

since then. In the year 2010, the frequency of high concentration days was about 68% of the days observed, while in 2005-2008 it was about 46 %. In the year 2012, the frequency of high concentration days was increased to 51%. For Mace Head, the Aitken mode $D_p$ had an opposite but a much weaker trend: there was an increasing trend in diameter before the year 2008, a decreasing trend from 2008-2010, and after that, the trend was increasing again. Based on this data set, we cannot derive the exact reasons for the changing $N$.



Figure 2. Number concentration of Aitken mode, Mace Head


**Figure 2.** DLM fit for Mace Head Aitken mode number concentration. Black dots represent daily averages of Aitken mode number concentrations in Mace Head. The red solid line represents the estimated level, and the red ribbon represents the 95% confidence interval for the level.





**Figure 3. Estimated trends for Aitken mode $D_p$ and $N$ at measurement sites. Trend has been calculated by DLM, see section 2.3.1 for details. The overall trend presented in the figure is comparable with the long-term trend estimates given in section 3.1. To get a DLM trend for one year, the one-day trend given by the model was multiplied by the number of days in a year (365 used for all years) and divided by the mean of the variable over the first observed year. For example, if the trend is showing an increase of 10% / year it means that if the short-term increase would continue for a year, the concentration would be increased by 10% during the year compared to the first year mean.**





In Fig. 3. we present the coefficients for in the DLM trend for Aitken mode $D_p$ and $N$. Mode parameters $D_p$ and $N$ were selected because those parameters are showing the strongest trends. Results for nucleation and accumulation modes are shown in the supplement (Figs S7 and S8). The trend derived using the DLM showed the transient changes in the level of the time series, in contrast to the constant, long-term trend estimated with the Sen-Theil estimator. The trend from the DLM was

constantly changing during the time series, achieving the best fit to the data as can be seen in Fig. 3. For Fig. 3., the unit of the change was scaled to be comparable with the long-term trends presented later. To get a DLM trend for one year, the one-day trend given by the model was multiplied by the number of days in a year (365 used for all years) and divided by the mean of the variable over the first observed year.

The most important result of the DLM analysis was that the trends are usually not monotonic during the measured period.

Therefore, long-term trends should be only thought of as approximation of the average change during the time period. It is also good to note that the mode parameters are connected, i.e., for some of the short-term trends observed in mode number concentration, there was an opposite trend in mode mean diameter. This can also be seen later in the long-term trends (Sen-Theil results) for some of the modes and sites.

The long-term trends were investigated using Sen-Theil estimators (Fig. 4). Number concentration $N$ of the modes showed the

largest changes over the investigated time periods, $D_p$ has the second largest changes, whereas $\sigma$ showed only minor variations compared to the other two parameters. This was similar for both Sen-Theil estimator and DLM results.

Amongst all variables and sites considered, accumulation mode $N$ showed the largest decrease, followed by Aitken and nucleation mode $N$ when long term trends are considered. Only urban sites showed consistent decreases in number concentration for almost all modes and sites. The only exception here is semi-urban Puijo that showed an increasing trend in

accumulation mode $N$. Urban sites are dominated by anthropogenic emissions (e.g., traffic and industrial activities), which are affected by recent air quality control measures in Europe. This naturally explains the decreasing trends in urban sites, as discussed in previous studies (Mikkonen et al., 2020; Sun et al., 2020). For rural and remote sites, there was more site-to-site variation in trends, and some of these sites showed trends of increasing $N$ in all three modes. The rural and remote sites are less directly affected by anthropogenic sources, but more by biogenic or other natural sources compared to urban sites. The

strength of the anthropogenic contribution varies between the rural and remote sites depending on the strength of the natural sources and transportation efficiency of air masses from more polluted environments. For example, the Central and Southern European rural sites are likely more affected by anthropogenic sources than Northern European rural or remote sites. The biogenic emissions depend greatly on environmental factors, which can vary significantly on a year-to-year basis and between sites. These factors may partly explain the large variation in trends between the different rural or remote sites. The difference

in trends of $N$ in the three modes at the same site may be related to different sources and their temporal changes. Furthermore, nucleation and Aitken mode particles are likely to be emitted or formed close to the measurement site, while accumulation mode particles are often transported to the location over longer distances. In particular, nucleation mode $N$ are dependent on the particle formation rate and their survival to larger sizes, which in turn are dependent on not only the precursor gas emissions





but also meteorological conditions and background particle concentrations (Nieminen et al., 2018). Thus, a decreasing trend
in the concentration of larger particles could even strengthen new particle formation.

Mace Head showed distinctly different behaviour compared to other sites as the number concentration of all three modes had increasing annual (Fig. 4) and seasonal trends (Fig. 5). It should be noted here that the investigated period of the Mace Head data set differs considerably from other investigated data sets: for Mace Head, the investigated period ends in the year 2012 while for other sites the time period ends in 2017 or 2018.

The long-term trends for the mode parameters showed that the decreasing trend in number concentration was not highly correlated with the increasing trend in $D_p$ for accumulation mode. Accumulation mode correlation between the estimated trend coefficients for $D_p$ and $N$ was -0.32. So, the decrease in number concentration was somewhat concurrent with increased particle size in accumulation mode. For the $\sigma$ parameter, the trend was almost zero for most of the sites.

For the Aitken mode and especially the nucleation modes, there were some sites that show an increase in $N$. For the Aitken
mode, the Spearman correlation between trend estimates of $D_p$ and $N$ was -0.31 and for nucleation mode, the spearman correlation was -0.48. Thus, in especially nucleation mode, some of the increases and decreases in number concentration were partially connected with a decrease or increase in $D_p$. Additionally, in nucleation and Aitken modes, $\sigma$ parameter showed only minor changes during the measured period.


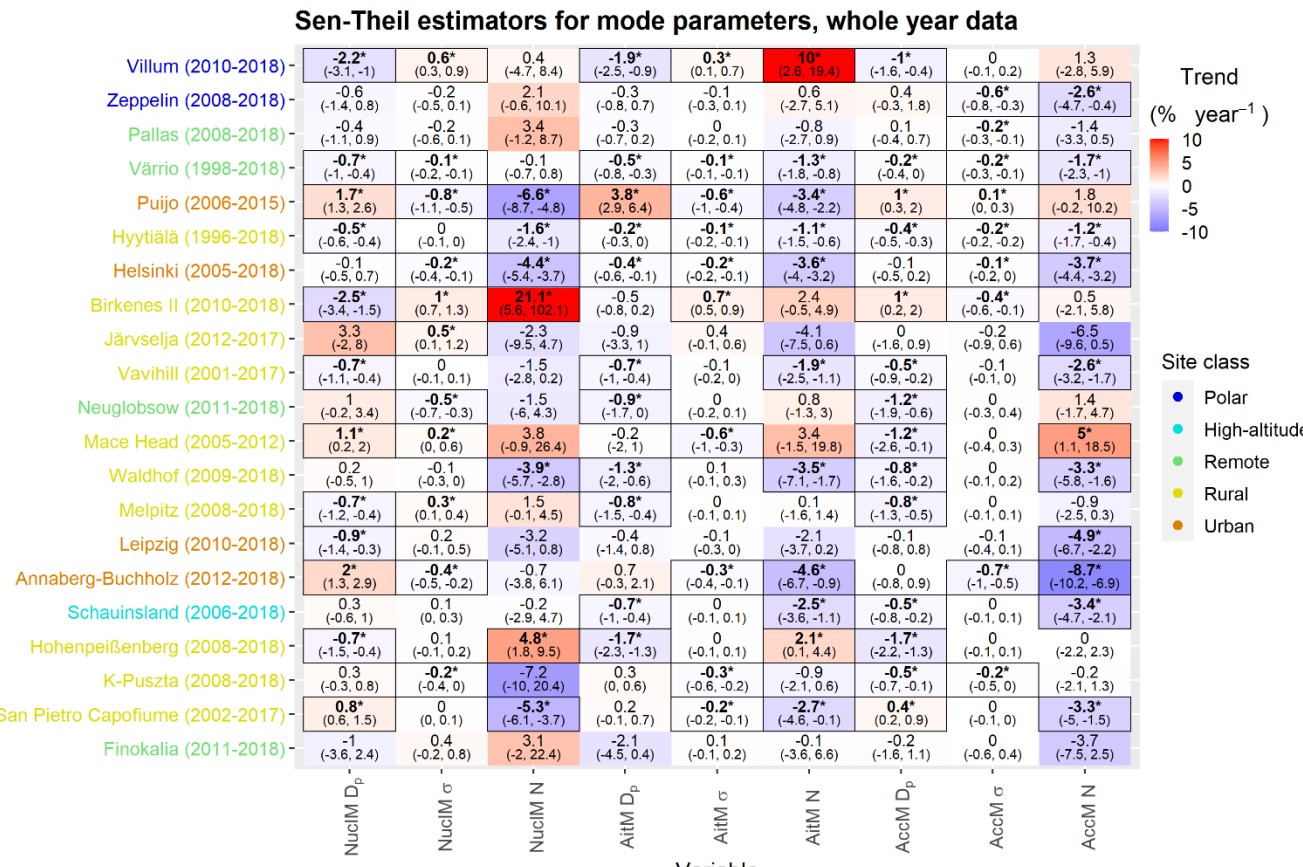

**Figure 4. Long-term trend estimators for measured trends of all mode parameters (mean geometric diameter $D_p$, geometric standard deviation σ, and number concentration $N$) in nucleation (NuclM), Aitken (AitM), and accumulation mode (AccM). Statistically significant (95% confidence level) trends are bolded, marked with an asterisk, and highlighted with border lines. Trends have been calculated using Sen-Theil estimator and complemented with bootstrap confidence intervals (see section 2.3.2).**

We also investigated if the trends have a seasonal behaviour. For seasonal trends in general, a decrease of $N$ was strongest for winter and weakest for summer (Fig. 5). In winter, there were relatively consistent decreasing trends all over Europe. In autumn (Fig. S9), the trends were also mostly decreasing. In summer and spring (Fig. S9), there were clear differences in trends between sites. Again, the most consistent trends were in urban sites, showing a decrease for accumulation and Aitken mode $N$. Nucleation mode $N$ for urban sites was also mostly decreasing. Other site classes were not showing consistent decreases possibly due to different contributions of anthropogenic and biogenic emissions between sites, discussed already earlier in this section. Sporadic large increasing trends in nucleation mode might be resulted from large portion of missing nucleation modes fitted and small concentrations which might cause large trends even for small absolute changes. During the winter season, this results in a stronger, decreasing trend in wintertime concentrations compared to summertime trends. This was most evident for accumulation and Aitken mode particles. Interestingly, especially during winter seasons, the nucleation mode exhibits an opposite observed trend than the accumulation and Aitken mode concentrations. As noted earlier, different trends in nucleation





mode number concentrations than for larger particles might be related to different sources and the effect of background particles on new particle formation acting as a condensation sink.



**Sen-Theil estimators for mode parameters, Winter**

**Sen-Theil estimators for mode parameters, Summer**



**Figure 5. Seasonal long-term trend estimates for all mode parameters (mean geometric diameter $D_p$, geometric standard deviation σ, and number concentration $N$ in nucleation (NuclM), Aitken (AitM), and accumulation mode (AccM) during winter (January, February, and December) and summer (June, July, and August). Statistically significant (95% confidence level) trends are bolded, marked with an asterisk, and highlighted with border lines. Trends have been calculated using Sen-Theil estimator and complemented with bootstrap confidence intervals (see section 2.3.2).**

### 3.2 Comparison of observed particle mode concentrations and climate model results

In this section, we compare the observational trends of $N$ of each mode to the trends of the climate model simulation data. These results are not fully comparable to the results presented in Section 3.1. since the investigated time period in this section is different from the time period in Section 3.1. For comparison of simulations and observations, at least seven years of data were required. Because model data was only available for the years 2001 through 2014, this limited the number of sites available for the comparison. Figures 6-8 display the thirteen sites that had sufficient data coverage for this time period. In the cases where measurement data was missing for a site for a certain month, model data for the corresponding month was omitted as well. As explained in Section 2.1.4, log-normal modes that were fitted to the measurement data were not directly comparable to the data provided by the climate models. We therefore additionally remapped the size distributions for specific size intervals (see section 2.1.4), which were used in the models, from the measurement data to correspond to the sectional (ECHAM-SALSA) and modal (EC-Earth3, ECHAM-M7, NorESM1.2, and UKESM1) representations of nucleation, Aitken, and accumulation mode as used in the models. To this end, we used the model-internal parameters to separate the respective modes (see Section 2.1.4 for details). In the following, we thus analysed three representations of the same measurement data, to which we will refer as "fitted modes" (Section 2.1.3) and "sectional" and "modal representation of the measurement data" (Section 2.1.4). While these three representations were not directly comparable to each other (because the size ranges for different modes varied between the different representations), it was still instructive to visualize them side by side. It should also be noted that the trends for the fitted modes in Figs. 6-8 were not the same as in Fig. 4, because the time intervals of the trend analyses were not the same.

### 3.2.1 Comparison of yearly trends

Figure 6 shows the trends in nucleation mode $N$. Unfortunately, at many measurement sites, the minimum detected particle diameter was too large to compute meaningful results for nucleation-mode-sized particles that were comparable to the models. Hence only five of the measurement sites (Hyytiälä, Helsinki, Vavihill, Melpitz, San Pietro Capofiume) could be compared to all models and three additional sites (K-Puszta, Pallas, Värriö) could be compared to models with modal aerosol representation. Of these sites, Hyytiälä, Helsinki, Vavihill, and San Pietro Capofiume showed comparable trends for all three representations of the measurement data, which were all decreasing and statistically significant. At all four of these measurement sites, the models showed decreasing trends as well, but in many cases, the negative trends were weaker and sometimes no significant trend was found. Observations at Pallas showed a strong increasing trend for both fitted mode and modal representation of the data, while all models showed slightly decreasing trends, of which two results were statistically significant.





When inter-comparing model results, we found that for most sites all models showed slight to medium decreasing trends (about 0 to -5% per year) for nucleation mode $N$. This was also expected, as all models used the same anthropogenic emission inventory, which exhibits a steadily decreasing trend in sulphur dioxide emissions over Europe for the modelled period (Hoesly et al., 2018). This directly affects nucleation rates and condensation rates of sulfuric acid in the models. There were only two
measurement sites that deviate from this general model trend. At K-Puszta, EC-Earth3 and ECHAM-SALSA showed increasing trends for the nucleation mode concentration. The other exception was a very strong decreasing trend in nucleation mode particle concentration for K-Puszta and Hohenpeißenberg in NorESM1.2. For both sites, however, the accumulation mode showed a positive trend in NorESM1.2, which was not present for the other models. A growing number of accumulation mode particles probably led to a larger condensation sink and therefore to suppression of new particle formation in the model.

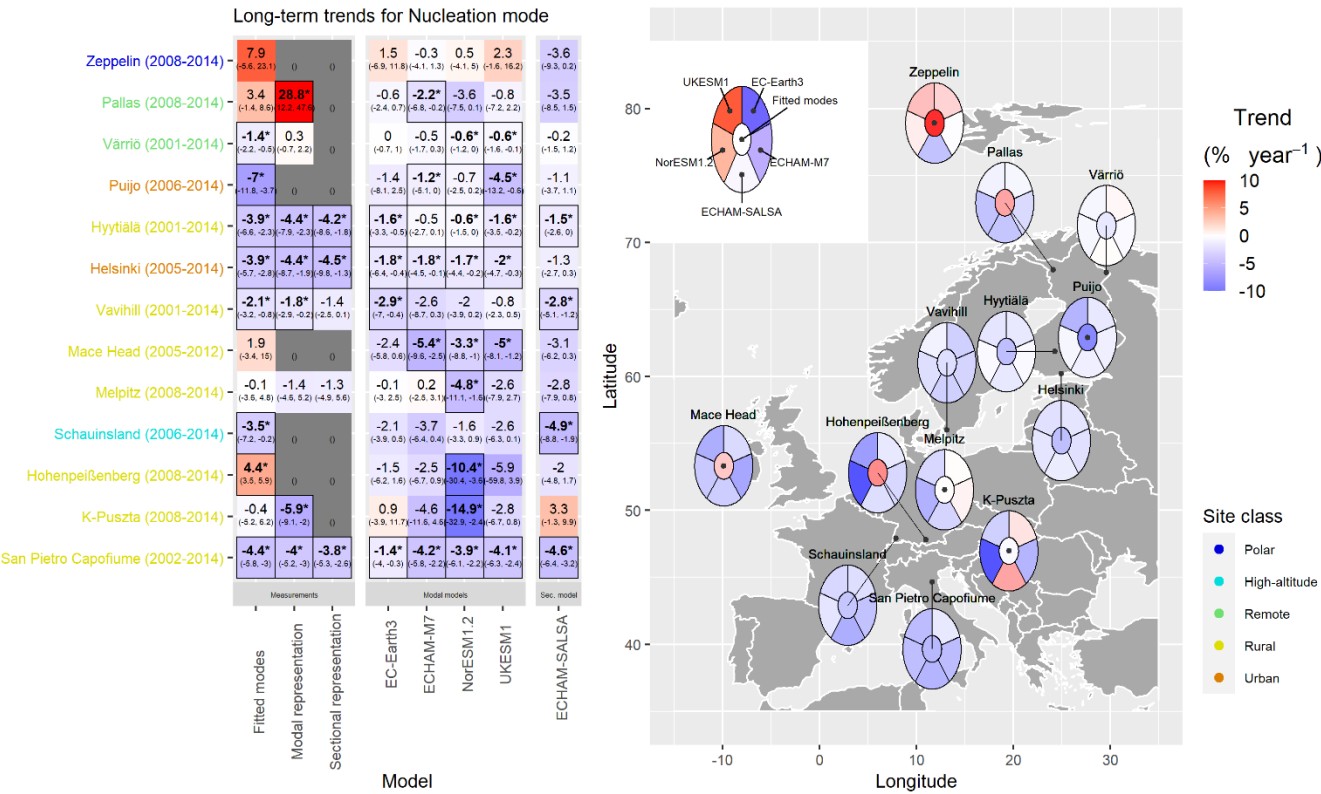


**Figure 6. Long-term trend estimates for measured and modelled nucleation mode number concentration. Left panel: Table of trends for different sites. The sites (y axis) are arranged from north to south. Statistically significant (95% confidence level) trends are bolded, marked with an asterisk, and surrounded by a black border. Right panel: Estimated trends presented in the map. The colour of the central part follows the trend of the fitted modes. Trends have been calculated using Sen-Theil estimator and complemented**
**with bootstrap confidence intervals (see section 2.3.2).**

Figure 7 shows the yearly trends in Aitken mode $N$. When the three representations of observations were investigated it can be concluded that the three different representations of the measurement data qualitatively agreed at most sites. The only exceptions were Pallas, where trends varied between –0.7 % (fitted mode) and +3.1 % per year (sectional representation), and



for Zeppelin, where the positive trend was weaker in the modal representation compared to the other two representations.
Furthermore, except for Zeppelin, Pallas, Mace Head and Melpitz, all observational trends for all three representations were statistically significant. Of all statistically significant trends, only Hohenpeißenberg showed a positive trend in Aitken mode *N* for all three observational representations. Mace Head and Zeppelin were quite different, as here the calculated trends for measurements were quite large and positive, but still not statistically significant. This is very likely explained by both sites' close vicinity to the ocean (O'Connor et al., 2008; Tunved et al., 2013).

When the trends in the models are investigated, most model trends at sites in Northern Europe were not statistically significant, while for the rest of the European sites, most trends were significant. Interestingly, the sectional model ECHAM-SALSA showed a significantly decreasing trend in most of the northern sites. This might be due to the different size limits used in the modal and sectional models. At most sites where both measurement and model trends were significant, the models agreed quite well with the measurements in both strength and direction of the trend. Though Hohenpeissenberg was an exception 610  where measurements showed a strong increasing trend, while the modelled trends were negative. The reasons for these differences are not clear.

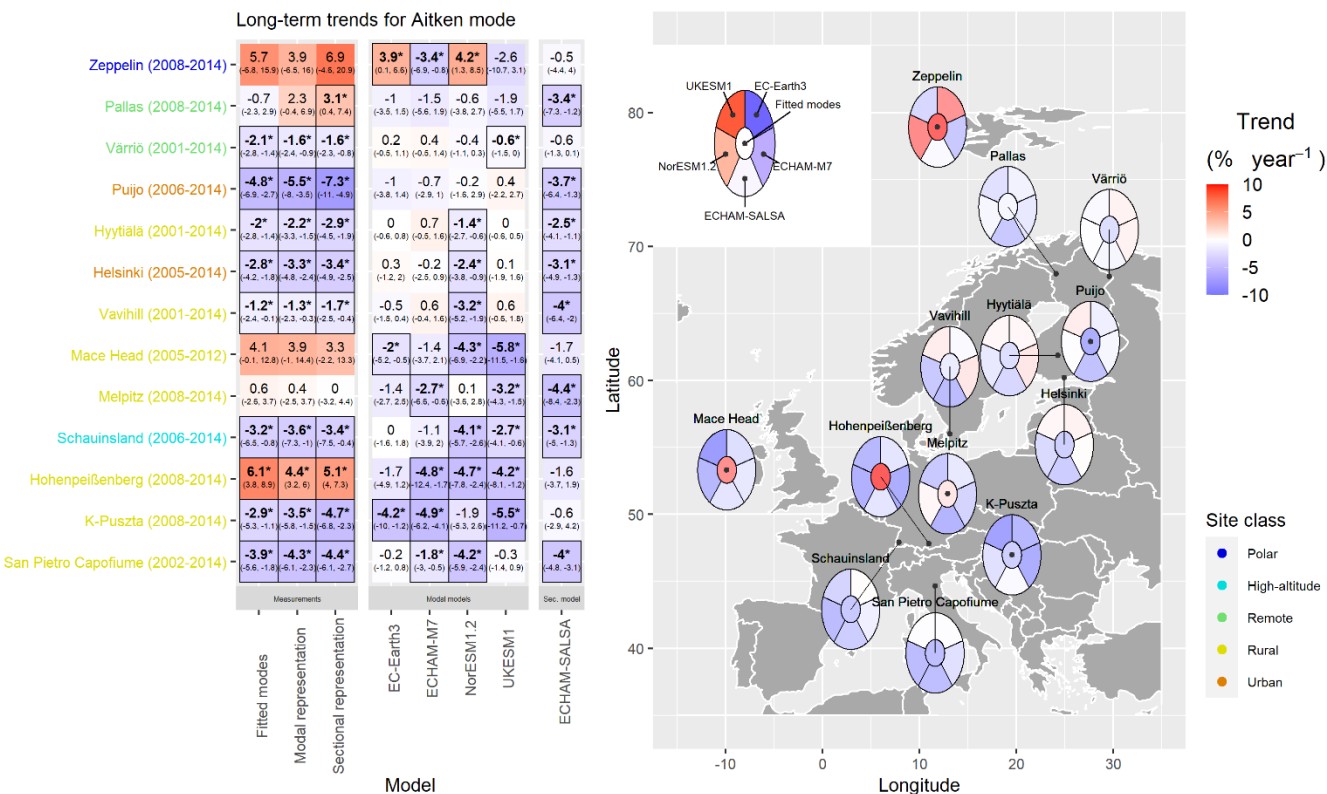

**Figure 7. Long-term trend estimates for measured and modelled Aitken mode number concentration. Left panel: Table of trends in different sites. Sites (y axis) are arranged from north to south. Statistically significant (95% confidence level) trends are bolded,**
**marked with an asterisk, and surrounded by a black border. Right panel: Estimated trends presented in the map. The colour of the central part follows the trend of the fitted modes. Trends have been calculated using Sen-Theil estimator and complemented with bootstrap confidence intervals (see section 2.3.2).**





Figure 8 shows the yearly trends in accumulation mode $N$. Again, for most measurement sites, the different representations of the measurement data showed statistically significant trends of equal direction and similar strength. Exceptions were Melpitz

and Hohenpeißenberg, which showed fairly weak, insignificant trends altogether, Zeppelin, which showed strong, opposite but, due to high variance, not statistically significant trends, and Puijo, which showed strong positive (but only partly significant trends) for all representations.

Concerning the model data, we did not find trends at any of the measurement sites that were statistically significant in the models. A general but weak tendency was, that occurrence of statistical significance increased with decreasing latitude of the

site. However, this tendency was not systematic in terms of which model produced significance at which site. Additionally, accumulation mode $N$ depend on wildfire, sea salt and mineral dust emissions and hence on the means of how these emissions are calculated and inserted into the model atmosphere. Considering these factors in combination with the relatively short period analysed here, a strong model internal and inter-model variability is to be expected.

There were only two sites, Helsinki and Vavihill, where all models and measurement representations agreed on the direction

of the trend (negative in both cases) in accumulation mode $N$. Some sites stood out because the different models found strong trends in opposite directions there. Hohenpeißenberg and K-Puszta stood out, as here the model trends were mainly negative except for NorESM1.2, which showed positive (albeit not significant) trends for both sites, as was also already discussed in connection with the nucleation mode trends.



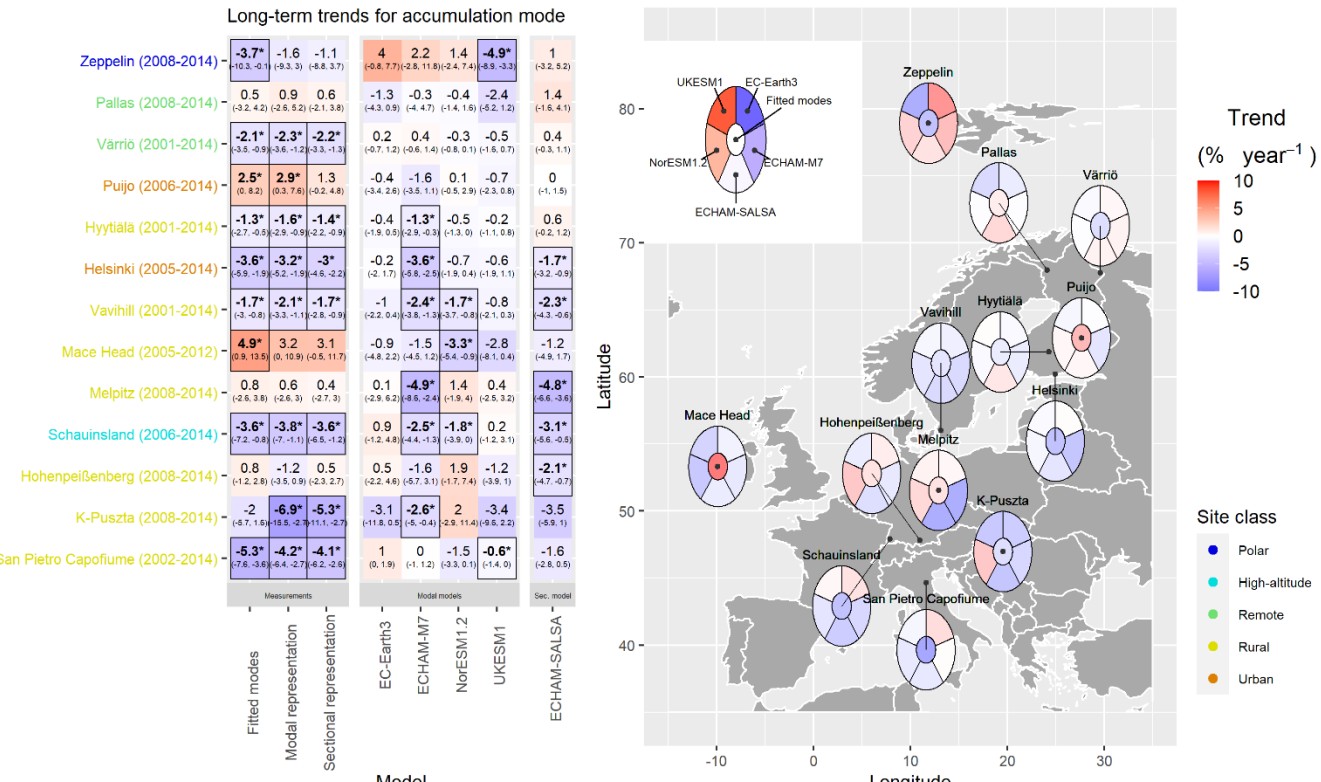

**Figure 8. Long-term trend estimates for measured and modelled accumulation mode number concentration. Left panel: Table of trends in different sites. Sites (y axis) are arranged from north to south. Statistically significant (95% confidence level) trends are bolded, marked with an asterisk, and surrounded by a black border. Right panel: Estimated trends presented in the map. The colour of the central part follows the trend of the fitted modes. Trends have been calculated using Sen-Theil estimator and complemented with bootstrap confidence intervals (see section 2.3.2).**

### 3.2.2 Comparison of seasonal trends

Seasonal trends of particle number concentration $N$ included more uncertainty than yearly average trends. Particularly the modelling results rarely showed statistically significant trends, even though the actual magnitudes of the calculated trends were often quite large. In general, the trends derived for the measurement data did not depend strongly on the representation used. There were few exceptions to this were Aitken mode trends at Zeppelin, Pallas, and Melpitz, and accumulation mode trends at Zeppelin, and Hohenpeißenberg. Seasonal model trends varied quite a lot between models, depending on the season, mode, and measurement site.

Figures 9 and 10 show the seasonal trends for Aitken and accumulation mode $N$, respectively, at all measurement sites analysed in Section 3.2.1. Unlike the results presented in Section 3.1 for the longer time series, the average decrease had been largest during spring and winter in Aitken and accumulation mode and during winter and spring in nucleation mode (Fig. S10), when comparing trends amongst all sites in monthly mode fitting data sets. There were also sites, such as Hohenpeißenberg and Mace Head, showing an increase for most of the $N$. Most of the sites were showing a decrease for all seasons.





Apart from a few exceptions, the measurements were showing decreasing seasonal trends of the Aitken mode $N$, which were also significant for some sites. The exceptions were Zeppelin and Mace Head. Additionally, the measurements at K-Puszta showed increasing trends in the autumn. In general, most of the significant model trends were negative and were found during
spring and summer. Neither observed or simulated data were showing significant trends in opposite directions for any of the two seasons, i.e., the significant seasonal trends were either decreasing or increasing for the one site and one measurement/model. Insignificant trends for the same site and measurement/model were sometimes decreasing for some seasons and increasing for some other seasons. The clearest difference between trends in modelled and measured data could be seen for the sites located in Finland, especially during winter and autumn where the measurements showed a decreasing
trend while the models mostly showed an increasing trend. Those differences observed during winter and autumn could affect the differences in yearly trends observed in Fig. 7.

There was no general agreement between different models concerning accumulation mode $N$ trends. The trends in the measurements for accumulation mode were mostly fairly similar to the Aitken mode trends. For many sites, these trends from measurements were significant only during spring. Aitken mode trends from models were mostly insignificant. As can be
expected from the yearly trends, the models reproduced measurement trends rather poorly, with no model performing much better or worse than any other model.





**Figure 9. Seasonal trend estimates for Aitken mode number concentration for four seasons: winter (Jan, Feb, Dec), spring (Mar, Apr, May), summer (Jun, Jul, Aug), and autumn (Sep, Oct, Nov). Sites are ordered from most northerly to most southerly. The bolded number, asterisk, and line border around the estimate indicate that the trend is statistically significant (95% confidence level). Trends have been calculated using Sen-Theil estimator and complemented with bootstrap confidence intervals (see section 2.3.2).**





**Figure 10.** Seasonal trend estimates for accumulation mode number concentration for four seasons: winter (Jan, Feb, Dec), spring (Mar, Apr, May), summer (Jun, Jul, Aug), and autumn (Sep, Oct, Nov). Sites are ordered from most northerly to most southerly. The bolded number, asterisk, and line border around the estimate indicate that the trend is statistically significant (95% confidence level). Trends have been calculated using Sen-Theil estimator and complemented with bootstrap confidence intervals (see section 2.3.2).



### 3.2.3 Comparison of seasonality and its pattern

In this section, we describe the seasonality and its pattern by the figures for nucleation (Fig. S17), Aitken (Fig. 11), and accumulation (Fig. 12) modes. More quantitative investigation based on SeasC and NIQR described in section 2.3.3 can be found in the Supplement, section S1.

When the modelled pattern of seasonality of observations and models are investigated, interesting differences and variations in the patterns were observed. For pattern of seasonality in modelled data, two models, NorESM1.2 and EC-Earth3, had

relatively consistent pattern for all sites, whereas for three models the seasonal cycle changed between north and south (Fig. 11 for Aitken mode and Fig. 12 for accumulation mode). NorESM1.2 and EC-Earth3 had relatively constant patterns of seasonality throughout Europe, even though the seasonal maximum variation between the sites varied. For NorESM1.2, nucleation mode had its maximum $N$ in winter (see Fig. S17), whereas Aitken and accumulation mode had their maximum $N$ in summer. EC-Earth3 had also consistent modes among all sites: nucleation mode had its maximum in summer, Aitken and

accumulation mode had their maximum in winter or early spring.

The other three models, ECHAM-M7, ECHAM-SALSA, and UKESM1, showed more clear changes in the patterns of seasonality between sites, typically showing stronger seasonality in northern sites. For Aitken mode, ECHAM-SALSA was showing two maxima in the seasonality; in Aitken mode is weaker in southern sites. ECHAM-SALSA showed also two maxima for nucleation mode (Fig. S17). ECHAM-M7 was showing the summer maximum for northern sites, whereas for

southern sites the seasonal curve was constant throughout the year or has the maximum in winter. Looking at the measurement-based representations (Modal and Sectional representation), the differences in seasonal patterns between the two ECHAM models were not only due to differences in Aitken mode diameter ranges. One likely contributor to the differences between M7 and SALSA was that they use different nucleation parameterizations. M7 uses the parameterization by Kazil et al. (2010) and SALSA uses the activation nucleation parameterizations by Sihto et al. (2006). In addition, it has been shown that since

solving simultaneously occurring nucleation and condensation within microphysical models will have implications on simulated new particle formation and growth of particles (Kokkola et al., 2009; Wan et al., 2013). Thus, the differences between M7 and SALSA are also related to differences in their numerical methods used for solving nucleation and condensation (see Kokkola et al., 2008, 2009). For the accumulation mode, these three models are showing a summer maximum at northern sites. For southern sites, ECHAM-SALSA is showing a summer maximum with a weaker seasonal

effect, and UKESM1 and ECHAM-M7 are showing consistent seasonal curves or winter $N$ maximums with weak seasonal effects. For nucleation mode, ECHAM-SALSA and ECHAM-M7 have two maxima in spring and autumn, whereas UKESM1 has typically only one maximum in winter or early spring (Fig. S17).

Additionally, modelled $N$s for different sites and the ratio between highest and lowest concentration sites varied significantly between the models. Differences in Aitken mode $N$s between models can be due to differences in model microphysics (see

Table 3) and especially in accumulation mode these differences can be due to varying deposition rates that affect the efficiency of long-range transportation of particles, or the way emissions are divided into different size ranges. Differences were large





especially in Aitken mode when we compared how $N$s were distributed between the sites in models and measurements. Furthermore, there were large variations in measured concentrations between the sites for all three investigated modes. The ratio for Aitken mode yearly median concentrations between highest and lowest concentration sites was between 65 and 90

for different measurement-based representations (Fitted modes, Modal and Sectional representation), and between 4 and 180 for models (see also Fig. 11). For Aitken mode, ECHAM-models had the least variation in concentrations between sites, followed by EC-Earth3, UKESM1, and NorESM1.2. For accumulation mode, ratios were smaller, between 34 and 40 for measurement-based representations and between 11 and 111 for models. For accumulation mode, the ratio for UKESM1, EC-Earth3, and ECHAM-M7 were between 11 and 15, 58 for ECHAM-SALSA, and 111 for NorESM1.2. A large difference

between ECHAM-models might be due to differences in accumulation mode diameters, and low concentration of accumulation mode particles in Zeppelin site in ECHAM-SALSA. The concentrations in sectional model representation (particle diameter 50-700 nm) were higher than for modal representation (100-1000 nm) for both ECHAM-models, and measurement-based representations.




**Figure 11.1 Seasonal cycle of Aitken mode number concentration in measurements and climate models for measurement sites. A subplot represents the seasonal cycle in one model or measurement. Coloured lines represent the median of the monthly means for Aitken mode number concentrations. Sites are ordered from most northerly to most southerly.**






**Figure 12. The seasonal cycle of accumulation mode number concentration in measurements and climate models for measurement sites. A subplot represents the seasonal cycle in one model or measurement. Coloured lines represent the median of the monthly means for accumulation mode number concentrations. Sites are ordered from most northerly to most southerly.**





## 4 Summary and conclusions

In this study, we had two aims: 1) to study the trends of particle modes, namely nucleation, Aitken, and accumulation, their properties ($N, D_p, \sigma$) in Europe and Arctic, and 2) to provide the first extensive comparison for climate model aerosol number concentration trends and seasonality with measured ones.

The results for measured data sets were in line with previous studies, showing that the number concentrations of particles were usually higher in urban sites and southern and Central Europe than in rural sites in northern Europe. Additionally, our results from measurements were showing a decreasing trend for most of the mode number concentrations and sites, which supports earlier findings. Our investigation for mode fittings revealed that mode diameter and number concentrations are dependent: increasing number concentration was sometimes related to a decrease in mode mean diameter. This dependency was stronger for particles of smaller diameters.

We also found that the trends in measured number concentrations have differences between seasons and that the decrease or increase was not constant during the time period. The Dynamic Linear Model (DLM) model was applied to characterize the changes of trends. DLM results supported our finding of dependence of diameter and number concentration in mode fitting data. In addition, we found that the changes in parameters are site-specific, i.e., time periods of decrease and at the same time increase among other sites of the same area were found. On the other hand, sites have distinct locations; if decreases would have been observed at the same time, it should have been resulted from uniform changes in the local particle properties at a larger area.

We compared measured and modelled trends for number concentrations. Before climate model comparison, the measured trends were made comparable with global model results by calculating corresponding sectional and modal representations also from the measured data. It was seen that the factors affecting the fitted modes, namely larger diameters in fitted modes and correlations between the mean diameter and number concentration, did not have a large role in the estimated trends from the measured data. Trend estimates for mode fitting data and corresponding sectional and modal representations were close to each other. For some sites, long-term measurements of small (< 10 nm) particles were not available, thus, conclusions about the nucleation mode trends for those sites were uncertain.

We found out that models were mostly able to reproduce long-term decreasing trends in Aitken and accumulation modes. Modelled trends of yearly data were usually smaller in absolute value but had the same direction than measured trend for most of the sites. However, for some of the sites, especially Mace Head and Hohenpeißenberg, were not similar when comparing measurements to climate models. We suspect that those sites are representing more local conditions than the area captured by the climate model grid box. For seasonal trends in general, the differences were larger. However, the number of data points in seasonal trend estimation is relatively small.

For seasonality representation, we found models having differences in their representation despite the emissions used in models being the same. There were differences in the seasonal pattern, its magnitude, and when the maxima of number concentrations are achieved. Furthermore, there were also differences in the model representation that comes to uniformity of the seasonal

pattern among the European continent: for ECHAM-M7, ECHAM-SALSA, and UKESM1, the seasonal pattern was varying between sites while for EC-Earth3 and NorESM1.2, the pattern was consistent for all sites. Also, the modelled number concentrations for different models had large differences. This could be potentially due to differences in the parametrizations of physical processes, which affects the model estimates of particle number concentrations in all size regimes. Our results indicate that availability and nature of observations we have, limits our ability to understand whether our models are accurately representing trends in particle concentrations and how this, in turn, affects ACI. We suggest that a more detailed characterization of processes causing model differences should be conducted in the future.

**Data and code availability**

Most of the particle number size distribution measurement data sets are already available from ACTRIS (https://actris.nilu.no/) and SmartSmear (https://smear.avaa.csc.fi/) databases. Data from Nieminen et al. (2018), missing measurement sites (Järvselja, San Pietro Capofiume, Villum), and model data as well as the codes are available upon request from authors.

**Author contribution**

HK, TYJ, TK, TN, AV, and SM planned the analysis, HK, TMie, TK, TB, KC, SD, MF, TH, NK, MK, AL, AM, NM, JPM, SMN, TvN, FMO, CO, DO, JBP, TP, ØS, MS, CS, HS, ES, TT, and AW participated in data collection, VL, SH, and TMii write the code for data analysis, VL, HK, TYJ, TMie, TK, TN, AV, and SM performed data analysis, analysed the results and contributed to the writing of the original draft with comments from all co-authors.

**Competing interests**

Some authors are members of the editorial board of journal Atmospheric Chemistry and Physics. The peer-review process was guided by an independent editor, and the authors have also no other competing interests to declare.

**Financial support**

This research has received funding from the European Union's Horizon 2020 research and innovation programme under grant agreements No 821205 (FORCeS) and 641816 (CRESCENDO). Academy of Finland Flagship funding (grant no. 337550, 337552 & 337549), the Academy of Finland competitive funding to strengthen university research profiles (PROFI) for the University of Eastern Finland (grant no. 325022). The research leading to these results has received funding from the European Union's Horizon 2020 research and innovation programme under grant agreement No 262254 (ACTRIS), No 654109 (ACTRIS-2), 739530 (ACTRIS-PPP), and 871115 (ACTRIS-IMP).



Tero Mielonen's and Harri Kokkola's work was supported by the Academy of Finland (grants No. 308292 and 317390).

Steffen M. Noe acknowledge funding from the Estonian Ministry of Sciences projects (grant nos. P180021, P180274, P200196), by the Estonian Research Infrastructures Roadmap Project "Estonian Environmental Observatory" (3.2.0304.11-0395). F. M. O'Connor was supported by the BEIS and DEFRA Met Office Hadley Centre Climate Programme (GA01101). C. E. Scott acknowledges funding from the UK's Natural Environment Research Council under NE/S015396/1. Erik Swietlicki acknowledges the support from the Swedish Research Council (Vetenskapsrådet) for ACTRIS Sweden under contract 2021-

00177. This research has been partly financially supported by the Danish Environmental Protection Agency and the Danish Energy Agency with means from MIKA/DANCEA funds for environmental support to the Arctic region (project nos. Danish EPA: MST-113-00-140; Ministry of Climate, Energy, and Utilities: 2018-3767) and ERA-PLANET (The European Network for observing our changing Planet) Projects; iGOSP and iCUPE, and finally by the Graduate School of Science and Technology, Aarhus University.

**Acknowledgements**

Villum Foundation is gratefully acknowledged for financing the establishment of Villum Research. Thanks to the Royal Danish Air Force and the Arctic Command for providing logistic support to the project. Christel Christoffersen, Bjarne Jensen, and Keld Mortensen are gratefully acknowledged for their technical support.

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
