# Peer review of "Comparison of particle number size distribution trends in ground measurements and climate models"

_Atmospheric Chemistry and Physics, 2022_

## Author Comment (AC1)

We gratefully thank both reviewers for the careful reading and valuable comments. Below we provide our point-by-point responses to the reviewers' comments. In the following context, raised comments/suggestions are marked in **black**, responses are presented in **red**, and changes to the manuscript/supplement information are indicated in **blue**.

In addition to the responses to the reviewers' comments, we have added a missing co-author (Radovan Krejci) and updated the Zeppelin dataset. The update of the dataset did not have major changes to the results. We also did some minor grammatical changes and some minor changes to the figures to enhance readability.

**Reply to Reviewer 1**

The authors perform a trend analysis using measured aerosol distributions modes (nucleation, Aitken and accumulation modes) over several long-term measurement sites across Europe and the Arctic. Trends are analyzed both seasonally and long-term (interannually). In addition, the trends derived from measurements are compared to those derived from models. While the trend analysis is thorough, the way in which results are presented are difficult to process as the authors rely on conveying the trends in several large tables. The coloring of the tables does help readability of the, but some of the discussed results would be better presented in more traditional direct comparison plots. Based on my comments shown below, I recommend publication after major revisions.

We would like to thank the reviewer for the constructive comments and several suggestions to improve the clearness of the presentation. We have now improved the figures according to comments and believe that this clarified the presentation. Below we give more detailed point-by-point responses to the comments.

General comments:

The manuscript is currently difficult to read. In general, the manuscript would benefit from some heavy editing. Additionally, the analysis uses measurements from 21 sites and a lot of site specific results are discussed and are somewhat meaningless without context or detailed site information. General results on measurement trends based on site class, latitude (north vs south) and agreement (or lack of agreement) with the models would provide a much clearer story.

We would like to thank the reviewer for these constructive comments. Instead of giving detailed site description in the manuscript, we have now added a reference for each site in table 1. In addition, we have added more detailed discussion on sites (section 2.1.1) used in model comparison as when the in-situ observation and large-scale models are compared, it is especially important to consider how representative the measurement sites are for the larger areas surrounding them.

"When the in-situ observations and large-scale models are compared, it is important to consider how representative the stations are for the larger areas surrounding them. The polar and remote sites (Zeppelin, Pallas, and Värriö) as well as rural site Hyytiälä can be considered to be representative for a larger regional fingerprint (Kyrö et al., 2014; Lohila et al., 2015; O'Connor et al., 2008; Tunved et al., 2013) and no large cities are locate close to these sites. It should be noted that the Värriö site can be impacted by pollution transported from the Kola Peninsula mining and industrial areas (200-300 km northeast from the station) at times (Kyrö et al., 2014). Mace Head represents marine environment excellently, when the airmasses arrive from Atlantic, but on the other hand can be affected by the continental outflow as well (O'Connor et al., 2008). The urban sites Helsinki and Puijo (as urban sites in general) are be affected by strong, local sources such as traffic or local industrial activity and the diurnal variation of the representativeness to the larger areas might be significant (Hussein et al., 2008; Leskinen et al., 2012). The rural (Hohenpeißenberg, K-Puszta, Melpitz, San Pietro Capofiume, Vavihill) sites represent European background well, but their representativeness for the model grid-box depends on the placement of the grid-box and on how large fraction of the grid-box is covered by large cities. It should be noted that Hohenpeißenberg is located at high altitude (988 m) and is classified as mountain site in some of the earlier studies (e.g. Rose et a. 2021) while Nieminen et al. (2018) classified it as rural site."

We agree with the reviewer, that the data set and comparison is complex, and would benefit from e.g. some scatter plots. We stumbled with several attempts and trials to produce clear plots with enough information and also to optimize the number of figures. We agree that the presentation is not perfect even if this type of presentation is used in earlier papers comparing large data sets and models. Unfortunately, the complexity of the data set is complicating the presentation. For example, we produced scatter plots to more quantify the trends both for observations and models, and for the differences between observations and models, for different site classes, for example. But unfortunately, the plots didn't bring any useful information to the analysis and manuscript (no clear trends or differences between site classes). We have now added these scatter plots (Figure S14-16) in the supplement and mention this clearly in the text (lines 709-710 in the updated manuscript).

"In general, the agreement between the trends of models and observations in trends of $N$ for all modes varied a lot within the site classes and no specific factor explaining the variation was found (see supplement figures S14-S16)."

Many of these "figures", are actually large tables that are difficult to digest. Figures 4 and 5, for example, have a lot of information, but are very difficult to extract trends in the... trends. While some cases have large trends, they are not necessarily significant which also makes it difficult to pick out consistent trends between sites that are also significant. I believe the site order is based on latitude, but I feel in some analysis it is more appropriate to group by site classification. The latitude doesn't really matter if pollution is the major influence on the site. Also the results would be much more informative if the trends were plotted against each other. For example, the authors discuss how N and Dp and other variables correlate. I suggest plotting these trends against each other and stick these tables in the supplement. Such comparisons might require having the site location being unknown, but you can at least color the points by site class. Another idea is, a bar graph showing N for one mode at each site would also make it easier to identify consistent trends and their magnitude. At the very least, I suggest removing sigma from these tables as there is little discussion on trends in sigma and trends in sigma are low in magnitude all around.

We thank the reviewer for suggesting several improvements to our figures. We have now tried to improve them accordingly. Below we give more detailed description of our improvements.

Figures 4 and 5: We changed the figures to bar plots showing the trends of N and Dp. We are showing confidence intervals and $\sigma$ in the supplement, using previous figures (Figures S9 for yearly data, Figures S11 and S12 for seasonal data).

Scatter plot of N and Dp was interesting to check, but as the dependency wasn't very clear and the figure did not bring essential new information, we decided to show that figure in the supplement (Figure S10).

Similarly for later figures 6-8 a direct comparison of measurements and model values would be easier to gauge than a large table of numbers. The tables have useful information for those that want site specific information and would be great in the supplement.

We changed the figures 6-8 to contain bar plots and maps. The tables are now moved to the supplement (Figure S13) to show exact numbers of trends. In the supplement we have now also added scatter plots of the measured and modelled values of the trend in N (Figs. S14-S16).

I wonder if the site classifications are appropriate. I am not familiar with many of the measurement sites and I am sure each has unique features; however, Mace Head is a well-known coastal site, that is greatly influenced by the Atlantic, but here it has been grouped together with rural sites despite this significant difference in aerosol source influence.

These site classifications are mostly based on Nieminen et al. (2018). We agree that the site classifications might not be perfect. As the reviewer pointed out, Mace Head does not belong to "rural" category and was unwittingly labeled as "rural" instead of 'remote' in most of the figures. Since we do not have a category for coastal or marine sites, it should be assigned as a "remote" site as already indicated in Table 1. This is now fixed in the figures. Even if we did not add a "marine" site class to our site classification, we have now brought up the specific character (Atlantic influence) of Mace Head site in the text (lines 176-177 in the updated manuscript).

The authors need to clearly indicate which figure is being discussing more often. Currently the authors have inserted the figures in the manuscript themselves in spots that make it relatively easy to identify the figure being discussed, but the figures are likely to be moved to different locations if accepted for publication.

We added more references to the figures throughout the results section.

Specific comments:

Line 38-39 – check sentence wording "in total of for".

We attempted to clarify the sentence and changed it to be "We investigated the trends and seasonality of particle number concentrations in nucleation, Aitken, and accumulation modes at 21 measurement sites in Europe and the Arctic."

Line 44-45 – reconsider the use of the word "stronger"

We replaced "stronger" by "higher".

Line 51 – I do not understand the use of the word "harmonized " in this context.

By "harmonized" we meant that there was same emission data as an input for the models. To clarify, we have modified the text to: "although all of the model simulations had identical input data to describe anthropogenic mass emissions".

Line 57 – "ACI altogether their ability to activate cloud droplets". This does not make sense.

Thank you for noting the typo. We have removed 'ACI' from the text.

Line 83 – incomplete sentence

Replaced one '.' with ','.

Lione 88 – While not ideal, it is straightforward on how to compare point measurements with models despite their differing temporal and spatial scale. You simply pick the grid box containing the measurement location and overlapping time range.  There is no other reasonable way.

We agree with the reviewer. We have modified the sentence to clarify our message:

"Interpretation and analysis of comparison of in-situ aerosol observations with global model outputs is not straightforward due to differing temporal and spatial scales represented."

Line 96 –do you mean co-locating?

Changed "collocating" to "co-locating".

Line 103 – section 2.0 (before section 2.1) is sort of an unnecessary summary of the rest of section 2. I suggest removing this text, but if you keep it at least references the relevant subsections in section 2 so that it is clear that you will go into more detail later.

We feel that such summary to introduce the methods may be useful for a reader as we are presenting a complex dataset and various different analysis. Therefore, we decided to keep this text and we added references to the relevant subsections.

Line 142-146 unclear. What is the "whole time period"? Why was the data interpolated?

By "whole time period" we mean here the whole measurement time period for a specific site that was included in the analysis in this study.

For those sites were detected size range changed over the measuring period, also the size resolution of the size distribution data varied. The data was interpolated for a site-specific size resolution in order to keep the further analysis consistent. We have clarified this by adding following sentence in the text: "Measurement data size bins were interpolated because otherwise, the size bins can vary during time series and hence, e.g. the calculated modal and sectional representations (see definitions from section 2.1.4) would be calculated from the different size bins."

Line 131- in line 110 you said 7 years of data.

7 years was the limit for measurement-model comparison (results section 3.2), 6 years were the limit for the measurement trend analysis (results 3.1). We highlighted the difference by modifying the sentence starting in line 139 in the revised manuscript: "For model comparison, in turn, we have included only those sites that have at least 7 years of a common time period with the model simulations (2001-2014) and sufficient data coverage (i.e., coverage > 50% of days)."

159-161 repetitive

We have revised the first sentence.

174 – what is "this size"?

Replaced "this size" with "the smallest detected size"

205 - "There were differences in nucleation mode representation during a day and during a year, nucleation mode most often being fitted after midday." What are the differences in representation?

By representation we meant the fraction between number of time points when modes were fitted and the total number of time points. We changed here 'representation' to 'coverage', being in line with Table 2, showing the coverages for the whole time period for each site.

208 - how would taking the mean affect the modes?

Thank you for pointing out this unclarity. The sentence was changed to "To conclude, the absence of modes did not drastically affect the daily mean of observed modes in Aitken and accumulation modes."

209 – why are nucleation mode number concentrations more uncertain?

Because fraction of fitted modes is less than for Aitken and accumulation mode. We added the clarification "As the fraction of fitted nucleation modes is smaller than for Aitken and accumulation modes," to the beginning of the sentence.

214 – a minimum of 5 days is quite a low limit for a monthly average.

While the limit of 5 days sounds quite low, for most of the months, number of days used for calculating average is much higher than 5. We tested to use 10 days as a limit instead of 5 days and noticed that the results didn't change significantly, however, the number of months used was (naturally) slightly lower. As the results were not very sensitive to the choice of this minimum limit, we set the limit of days to 5 in order to have more months included in the analysis.

251 – table 3?

Corrected.

327-329 "We used the Dynamic linear model (DLM)" is stated twice in the same sentence.

Corrected

363 – what is "seasonal data of monthly averages"?

It means monthly averages of a specific season, e.g. winter months (December, January, February) only. To clarify our unclear expression, we changed it to "monthly averages of a specific season".

439 - You do not label any sites as "mountain" or "continental" so it's not totally clear how you can make this comparison. It's also not clear what makes a site "continental" vs urban or rural.

It is true that the site classification can vary between different studies. Unlike the classification used in our study, Rose et al. (2021) used "mountain" and "continental" site classes for example. Hence, even if there are some same sites used in Rose et al. and our study, the classification was different. (Rose et al., 2021) We have now clarified that in the text and bring up clearly the stations used in our and Rose et al. study as well the corresponding classification in both studies.

"Results for $D_p$ are somewhat different compared to what has been observed in Rose et al. (2021). Rose et al. (2021) used a slightly different site classification than employed in this study. Unlike the classification used in our study, they classified the stations both based on geographic area (e.g. mountain and continental site classes) and based on footprint (e.g. urban and rural site classes). In their study, one site could have belonged to more than one site class. Hence, even if there are same sites used in Rose et al. and our study, the classification was different. With their classification, they reported that mode diameters for Aitken and accumulation modes were smallest for urban sites (32 ± 11 nm and 122 ± 37 nm, Leipzig in our study), followed by mountain (39 ± 9 and 142 ± 25, Hohenpeißenberg), polar (42 ± 14 and 149 ± 37, Pallas, Värriö, and Zeppelin), and continental (51 ± 13 and 174 ± 29, Annaberg-Buchholz, Birkenes II, Hyytiälä, K-Puszta, Leipzig, Melpitz, Neuglobsow, Schauinsland, Vavihill, and Waldhof) sites. (The sites used in both studies are mentioned in the brackets)."

450 – As you mentioned, you are limited by your measurement lowest diameter. Also, the nucleation mode starts small and grows via condensation, so the measurements likely simply occurred after some growth of the nucleation mode.

We added a sentence "Higher nucleation mode mean diameter detected in the measurements may be due to the lowest detectable diameter being usually around the upper limit of model representations. As the measurement do not capture the smallest nucleated particles and only detect them after some growth, the average nucleation mode diameters determined from measurements may be an underestimation." to clarify this.

484 – Hold off on mentioning the sen-theil estimator until discussing it with Figure 4.

We have now removed the mentioning and deleted ", in contrast to the constant, long-term trend estimated with the Sen-Theil estimator" from the text.

506-507 how do you know this? Source?

Lines 506-507 refers to "For example, the Central and Southern European rural sites are likely more affected by anthropogenic sources than Northern European rural or remote sites". The sentence was meant to be speculative. We have now given also the reason for our speculations and modified the text:

"For example, the Central and Southern European rural sites are likely more affected by anthropogenic sources than Northern European rural or remote sites due to denser incidence of large urban areas in Central and Southern Europe."

518 why does the mace head site data end in 2012? I thought this site was still making measurements.

In Actris database, there are data until 2012 so the later data weren't available from there for this study.

520 this sentence is redundant with the next sentence and somewhat contradicts the sentence starting on 522.

First sentence deleted.

579-580 only 1 model shows a statistically significant trend for the Pallas nucleation mode.

Thank you for pointing this out. We have corrected this in the text.

605 "When the trends in the models are investigated" this is unnecessary text...

Removed.

Line 608-609 Plotting measurement vs model trend would really help gauge your argument. Furthermore, is it even meaningful that only the statistically significant values agree?

As mentioned in earlier comments, we have now added a measurement vs model trend scatter plot to the supplement (Fig. S14-S16). In addition, we added bar plots of trends to Figures 6-8 for better visualization.

626- A major contributor to accumulation mode particles is cloud processing.

We changed the sentence to "Additionally, accumulation mode $N$ depends on wildfire, sea salt, and mineral dust emissions (and atmospheric processes such as cloud processing), and hence on the means of how these emissions are calculated and inserted into the model atmosphere."

644- reference the figure before discussing the results in the figure.

We replaced the first sentence of the second paragraph to a first sentence of a first paragraph.

683 "When the modelled pattern of seasonality of observations and models are investigated, interesting differences and variations in the patterns were observed." Only when being investigated? Unnecessary text.

Removed.

750 – I do not understand what is being stated here. Of course sites have distinct locations. Local particle properties at a larger area? What larger area?

The referee is correct, our initial sentence was unclear. We have now clarified:

"On the other hand, sites are considered as point measurements; which means that if decreases would have been observed at the same time in a certain area, it should have been resulted from uniform changes in the particle properties at a regional level."

The introduction is a little hard to follow. Better transition words/sentences would greatly help the flow of the text.

We have improved the readability of the introduction by adding transition sentences/words.

Section 2 – Please provide definitions of the different site types, particularly those that are not commonly used (rural vs rural regional background)

We changed the site classes to a commonly used classes (polar, high-altitude, remote, rural, and urban) following Nieminen et al. (2018). We have now mentioned this in the manuscript text (lines 158-163 in the updated manuscript.

"In this study we use commonly used site classes (polar, high-altitude, remote, rural and urban) following Nieminen et al. (2018). Site environment classification is adapted from Nieminen et al. (2018) for those sites that were included in their study. For other sites, we have used classifications from the literature (Sun et al., (2020) for German sites, Leskinen et al. (2012) for Puijo, Schmale et al. (2018) for Vavihill, and Nguyen et al. (2016) for Villum) for environment classification and adjusted their classification according to Nieminen et al. (2018). The detailed description of each site, including the facility and environment descriptions, can be found in literature (see Table 1)."

Because the current site classification is commonly used, we decided to not add the definitions of site types into the manuscript.

Section 2.1.2 is vague. You should provide all ways in which the data was quality checked for reproducibility, not a single example.

Some parts of the quality check routines of mode fitting were mentioned later in section 2.1.3 instead of discussing them in 2.1.2 to keep the chronological order of the details discussed. We decided to merge the section 2.1.2 with the section 2.1.3 because, as pointed out by the reviewer, the section 2.1.2 described the performed quality check inadequately.

Figure 6-8 I suggest coloring the text in the map using your site class as well.

Fixed.

**Reply to reviewer 2**

The current manuscript is a sound and can be accepted after minor revision. I congratulate the authors about all the details, I do not doupt about the quality of the data and the validation and comparison of the models.

I would suggest to write the abstract to a much much user friedly way. How do the models compare? What is needed to improve them? What is the current limitation and wha is needed to improve modelling of particle number size distributions? I am not sure reading the manuscript I understood that.

The authors did a marvellous job collecting this huge amount of data, please consider selling the message to non modellers

We would like to thank the reviewer for the positive comments and improvement suggestions. We have modified the manuscript accordingly.

We modified the abstract, as well as the conclusions section, and added some sentences related to the questions above. In short, we suggest that differences are most likely due to complex effect of different processes, instead of one specific feature (e.g. aerosol representation or emission size) or process of models. Based on our results, differences in model microphysics, especially aerosol deposition rates and efficiency of long-range transportation, are suggested as topics of more in detail investigation in future studies.

We hope these changes also help to convey some of the main points of study to non-modellers. Particularly this in mind, we also added couple sentences in the conclusions. In the first paragraph we added: "In addition to providing a dataset for model evaluation, the observational data compiled in this study could also facilitate studies on how the aerosol size distributions have evolved during previous years and how it has changed, e.g., the cloud activation capability of aerosol."

In the last paragraph we added: "In addition to consistent long-term data, good characterization of the measurement sites and the surrounding areas that they present is important for a thorough comparison between models and observations."

**References**

[revised manuscript text omitted]

---

## Author Response (AR2)

We thank the reviewer for the careful reading and valuable comments. Below we provide our point-by-point responses to the reviewer's comments. In the following context, raised comments/suggestions are marked in black, responses are presented in red, and changes to the manuscript are indicated in blue.

**Reply to Reviewer 1**

I want to commend the authors very thorough effort to improve the manuscript and respond to reviewer's comments. I believe the manuscript has greatly improved in quality. I have only a few minor comments listed below.

Note, line numbers are from the manuscript with track changes.

We thank the reviewer for the positive comments.

Line 45 "….the effect of aerosols has the largest uncertainty in global climate model radiative forcing estimates." Largest uncertainty relative to? Maybe add something like "relative to other drivers of radiative forcing" to the end of the sentence.

We clarified that we indeed meant to compare the drivers of radiative forcing and changed the sentence to "Despite a large number of studies, out of all drivers of radiative forcing, the effect of aerosols has the largest uncertainty in global climate model estimates."

Line 491-494 You repeat yourself in this sentence.

Thank you for noticing this. We removed the end of sentence ("and "continental" site classes for example, and classified the stations both based on geographic area and based on footprint").

Line 501 "However, otherwise,..." These words have similar meaning. I would remove one.

We removed "However,".

Line 835 – 837 Please revise: "...if decreases would have been observed at the same in a certain area..."

We revised it to be "…if decreases in the particle properties would have been observed at the same time in a certain area…".